# A Bayesian and efficient observer model explains concurrent attractive and repulsive history biases in visual perception

Matthias Fritsche, Eelke Spaak, Floris P de Lange*

Donders Institute for Brain, Cognition and Behaviour, Radboud University, Kapittelweg, Netherlands

**Abstract** Human perceptual decisions can be repelled away from (repulsive adaptation) or attracted towards recent visual experience (attractive serial dependence). It is currently unclear whether and how these repulsive and attractive biases interact during visual processing and what computational principles underlie these history dependencies. Here we disentangle repulsive and attractive biases by exploring their respective timescales. We find that perceptual decisions are concurrently attracted towards the short-term perceptual history and repelled from stimuli experienced up to minutes into the past. The temporal pattern of short-term attraction and long-term repulsion cannot be captured by an ideal Bayesian observer model alone. Instead, it is well captured by an ideal observer model with efficient encoding and Bayesian decoding of visual information in a slowly changing environment. Concurrent attractive and repulsive history biases in perceptual decisions may thus be the consequence of the need for visual processing to simultaneously satisfy constraints of efficiency and stability.

## Introduction

A fundamental question of human visual perception is how our brains efficiently and accurately process the flood of visual information that continuously bombards our retinae. Importantly, our visual environment exhibits strong temporal regularities (*Dong and Atick, 1995*; *Schwartz et al., 2007*; *Simoncelli and Olshausen, 2001*); for instance, in natural viewing behavior, orientation information tends to be preserved across successive time points and thus stable over short time scales (*Felsen et al., 2005*; *van Bergen and Jehee, 2019*). Our brains could potentially exploit this temporal stability and leverage information from the recent past to optimize processing of new visual input.

Indeed, there is ample evidence that our brains adapt visual processing in response to recent experience. For example, a prolonged exposure to a stimulus typically induces a repulsive bias in the percept of a subsequent stimulus – a phenomenon termed repulsive adaptation. Repulsive adaptation has been reported for a wide range of visual features such as orientation (*Gibson and Radner, 1937*; *Jin et al., 2005*), color (*Webster and Mollon, 1991*), and numerosity (*Burr and Ross, 2008*). Prominent theories of adaptation posit that repulsive biases may be explained by an optimally efficient encoding of visual information, given temporal regularities in the input. Mechanisms underlying such efficient encoding could include self-calibration and decorrelation (*Barlow and Földiák, 1989*; *Clifford et al., 2000*; *Müller et al., 1999*), or an increase in the signal-to-noise ratio for measurements similar to recently perceived stimuli (*Stocker and Simoncelli, 2006*). Such a mechanism would increase sensitivity to small changes in the environment (*Clifford et al., 2001*; *Mattar et al., 2018*; *Regan and Beverley, 1985*).

*For correspondence:
floris.delange@donders.ru.nl

In contrast to repulsive adaptation, recent studies have found that perceptual decisions about stimulus features can be attracted *towards* stimuli encountered in the recent past – a phenomenon termed attractive serial dependence (*Fischer and Whitney, 2014*). Like repulsive adaptation, attractive serial dependence has now been reported for a large variety of visual features, such as orientation (*Cicchini et al., 2017*; *Czoschke et al., 2019*; *Fritsche et al., 2017*), numerosity (*Cicchini et al., 2014*; *Corbett et al., 2011*; *Fornaciai and Park, 2018*) and spatial location (*Bliss et al., 2017*; *Manassi et al., 2018*; *Papadimitriou et al., 2015*). This phenomenon may represent a (Bayes-)optimal strategy of decoding the sensory information into a perceptual decision (*Cicchini et al., 2018*; *van Bergen and Jehee, 2019*). Effectively, attractive serial dependence may smooth perceptual representations over time, in order to promote visual stability in the face of disruptive factors, such as eye blinks and external or internal noise. Although it is evident that both repulsive and attractive biases occur (*Fritsche et al., 2017*), it is currently unclear whether and how these biases interact during perceptual decision-making.

If repulsive adaptation and attractive serial dependence reflect the distinct mechanisms of encoding and decoding, they may have separate timescales that are optimized for their respective process. Previous studies on attractive serial dependence suggest that the perceptual history attracts subsequent perceptual decisions over a relatively short time period of about 10–15 s (*Fischer and Whitney, 2014*; *Van der Burg et al., 2019*). Reports on the timescale of repulsive adaptation are more variable. While some studies have found that adaptation in response to brief stimuli is short-lived, vanishing after a few seconds (*Kanai and Verstraten, 2005*; *Pastukhov and Braun, 2013*; *Pavan et al., 2012*), recent studies have reported minutes-long repulsive biases in perceptual decisions (*Chopin and Mamassian, 2012*; *Gekas et al., 2019*; *Suárez-Pinilla et al., 2018*). Together, these studies raise the possibility that attractive serial dependence and repulsive adaptation biases may occur over dissociable timescales. This could provide an opportunity for disentangling history-dependent encoding and decoding mechanisms underlying perceptual decision-making, even within the same perceptual decision.

In the current study, we examined the timescales over which current visual processing is attracted and repelled by the perceptual history. Across four experiments, we show that attraction and repulsion have distinct time scales: perceptual decisions about orientation are attracted towards recently perceived stimuli, but repelled from stimuli that were experienced further in the past. Importantly, we demonstrate that the long-term repulsive bias is spatially specific, akin to classical sensory adaptation (*Boi et al., 2011*; *Knapen et al., 2010*; *Mathôt and Theeuwes, 2013*) and in contrast to short-term attractive serial dependence (*Fritsche et al., 2017*). Furthermore, it exhibits a long exponential decay, with stimuli experienced minutes into the past still exerting repulsive effects. Moreover, we find that this pattern of concurrent attractive and repulsive biases cannot be explained by a Bayesian ideal observer model, which exploits the temporal stability of the environment (*van Bergen and Jehee, 2019*). We present a novel ideal observer model, in which repulsive biases arise due to efficient encoding (*Stocker and Simoncelli, 2006*; *Wei and Stocker, 2015*), and attractive biases are due to Bayesian decoding of sensory information. This model illustrates a principled way in which observers can exploit the temporal stability in the environment across multiple processing stages to concurrently achieve efficient and stable visual processing.

## Results

### Experiment 1: Perceptual decisions are attracted towards short-term history but repelled from long-term history

The aim of the first experiment was to characterize the dependence of perceptual decisions on the short- and long-term stimulus history. To this end, twenty-three human observes completed a series of trials, in which they were presented with oriented Gabor stimuli, randomly sampled on each trial from the full range of possible orientations (0, 180°], and subsequently had to report the perceived orientation of each stimulus by adjusting a response bar (*Figure 1*). We then analyzed the dependence of the current adjustment response on each stimulus experienced in the 40 preceding trials, corresponding to a time window reaching ~3.5 min into the past. We found that adjustment responses were systematically attracted towards the stimulus presented on the previous trial (p<0.0001; *Figure 2A*). This attraction bias was maximal when orientations of successive stimuli

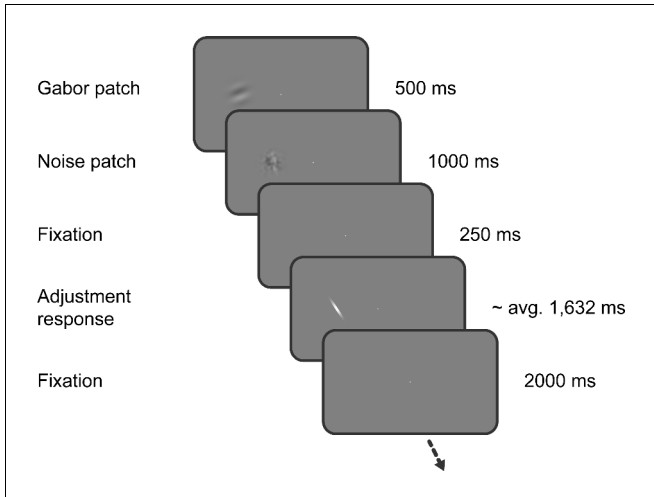

**Figure 1.** Task of Experiment 1. Observers saw a Gabor stimulus followed by a noise mask and subsequently reproduced the orientation of the stimulus by adjusting a response bar. Stimulus presentation in the left or right visual field alternated between separate, interleaved blocks.

differed by 18° (max. attraction = 1.66°), similar to previous reports of attractive serial dependence (*Fischer and Whitney, 2014*; *Fritsche et al., 2017*). However, the attractive influence of previous stimuli was short-lived and was only clearly observable between adjacent trials, corresponding to an inter-stimulus interval of about 5 s (*Figure 2C*). Conversely, stimuli encountered further back in the past exerted a repulsive bias on the current adjustment response. The repulsive bias was maximal for stimuli presented four trials in the past (*Figure 2B*, p<0.0001, max. repulsion = −1.07°) and subsequently decreased towards zero, albeit remaining negative up to 22 trials back, corresponding to a time window of about two minutes. Similar to the short-term attractive bias, the repulsive bias was tuned to the orientation difference between the current and past stimulus, such that current adjustment responses were maximally repelled for orientation differences of ~19° (average tuning width of 4- to 22-back trials). The results of Experiment 1 thus indicate that perceptual estimates are subject to both attractive and repulsive biases, operating over different timescales. While perceptual estimates are attracted towards immediately preceding stimuli, they are repelled from temporally distant stimuli. This repulsive bias potentially reaches back minutes into the past. The orientation tuning of the long-term repulsive bias is reminiscent of sensory adaptation (i.e. the negative tilt-aftereffect), which typically induces maximal repulsion for orientation differences of about 20° (*Gibson and Radner, 1937*; *Jin et al., 2005*). To shed more light on the nature of the long-term repulsive bias, we next investigated whether it is modulated by the spatial overlap between the current and temporally distant stimulus. Such a spatial specificity is a hallmark of classical sensory adaptation effects (*Boi et al., 2011*; *Knapen et al., 2010*; *Mathôt and Theeuwes, 2013*), and thus would point towards low-level sensory encoding as the origin of the repulsive bias.

## Experiment 2: Long-term repulsive biases are spatially specific

Experiment 2 was similar to Experiment 1, with the exception that stimulus position could vary between two locations (10 visual degrees apart) on a trial-by-trial basis. Consequently, any past stimulus could be presented at the same or a different spatial location as the current stimulus. This dataset, comprising 24 human observers, has been previously described (*Fritsche et al., 2017*). We previously showed that adjustment responses were systematically biased towards the immediately preceding stimulus, with a similar magnitude when it was presented at the same or a different location (see Experiment 1 in *Fritsche et al., 2017* and *Figure 3A and B*; same location: max. attraction = 1.15°, p=0.005; different location: max. attraction = 1.17°, p=0.0006; same vs. different: p=0.97). Here, we sought to investigate the spatial specificity of the long-term repulsive bias. To this end, we focused on the past 10 stimuli preceding the current stimulus, for which repulsive biases in Experiment 1 were strongest. In line with Experiment 1, we found that adjustment responses were

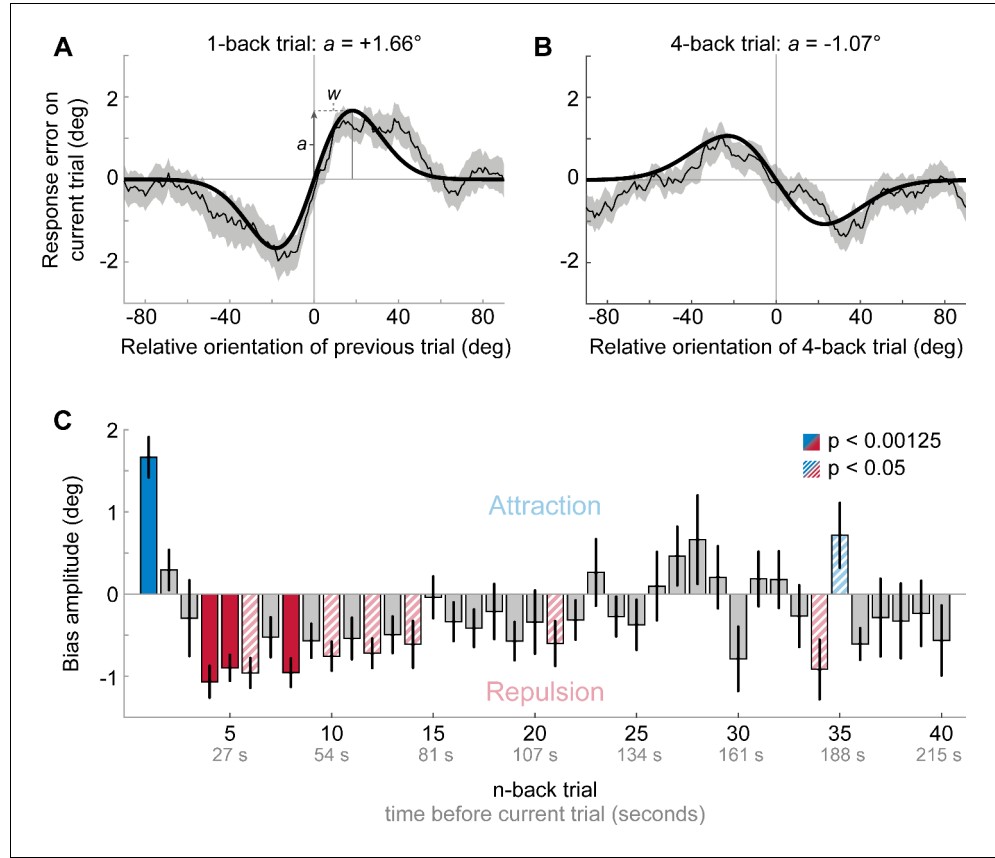

**Figure 2.** Results of Experiment 1: Estimation responses are attracted towards short-term, but repelled from long-term stimulus history. (**A**) Serial dependence of current response errors on the previous stimulus orientation. We expressed the response errors (y-axis) as a function of the difference between previous and current stimulus orientation (x-axis). For positive x-values, the previous stimulus was oriented more clockwise than the current stimulus and for positive y-values the current response error was in the clockwise direction. Responses are systematically attracted towards the previous stimulus, as is revealed by the group moving average of response errors (thin black line). The attraction bias follows a Derivative-of-Gaussian shape (DoG, model fit shown as thick black line). Parameters $a$ and $w$ determine the height and width of the DoG curve, respectively. Parameter $a$ was taken as the strength of serial dependence, as it indicates how much the response to the current stimulus orientation was biased towards or away from a previous stimulus with the maximally effective orientation difference between stimuli. Positive values for $a$ mark an attractive bias. Shaded region depicts the SEM of the group moving average. (**B**) Current responses are systematically biased away from 4-back stimulus. (**C**) Attraction and repulsion biases exerted by the 40 preceding stimuli. Bias amplitudes show the amplitude parameter $a$ of the DoG models, fit to the n-back conditioned response errors (see panel **A and B**). While the current response is attracted towards the 1-back stimulus, it is repelled from stimuli encountered further in the past. Colored bars indicate significant attraction (blue) and repulsion (red) biases (solid: Bonferroni correction for multiple comparisons; striped: no multiple comparison correction). Error bars represent 1 SD of the bootstrap distribution. The online version of this article includes the following source data for figure 2:

**Source data 1.** Results of Experiment 1: Estimation responses are attracted towards short-term, but repelled from long-term stimulus history.

significantly repelled away from stimuli seen 4 to 9 trials ago (*Figure 3A*). Moreover, stimuli presented within this time window, reaching from 26 to 60 s in the past, exerted a spatially specific bias on the current adjustment response. That is, the current adjustment response was more strongly repelled by a previous stimulus presented at the same compared to a different spatial location as the current stimulus (*Figure 3C*; same location: max. repulsion = −0.81°, p<0.0001; different location: max. repulsion = −0.39°, p=0.0006; same vs. different: p=0.006). This spatial specificity of the repulsive bias, exerted by stimuli seen tens of seconds in the past, points towards a sensory origin of the long-term repulsive bias, akin to the negative tilt-aftereffect. Moreover, the small remaining

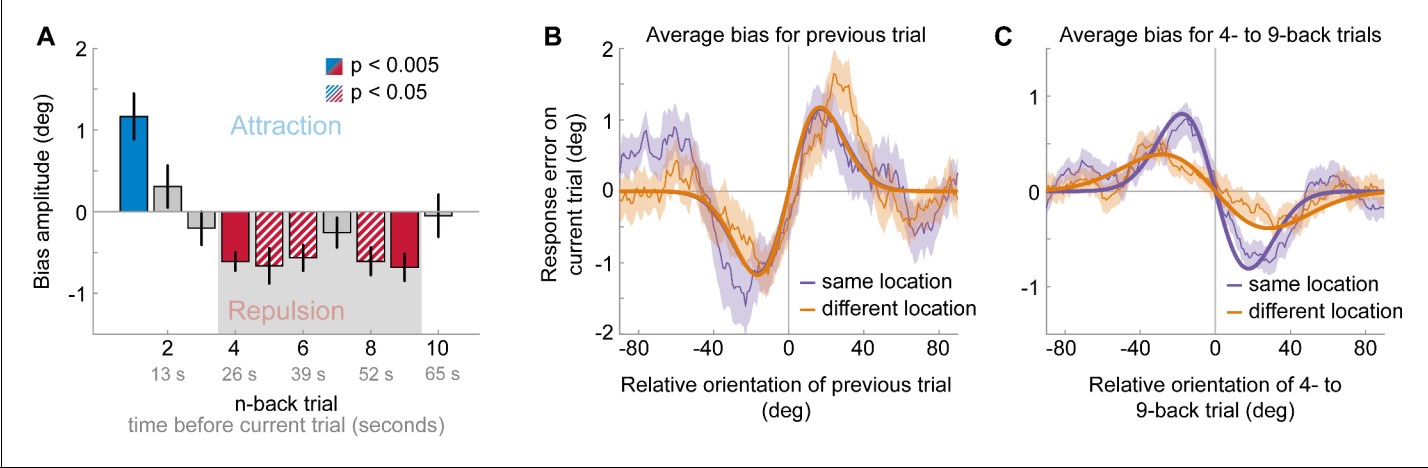

**Figure 3.** Results of Experiment 2: Long-term repulsive biases are spatially specific. (**A**) Attraction and repulsion biases exerted by the 10 preceding stimuli, regardless of changes in spatial locations. The current response is attracted towards the previous stimulus, but repelled from stimuli seen 4 to 9 trials ago. Colored bars indicate significant attraction (blue) and repulsion (red) biases (solid: Bonferroni correction for multiple comparisons; striped: no multiple comparison correction). Error bars represent 1 SD of the bootstrap distribution. The gray shaded area marks the time window of interest for which we estimated the spatial specificity of the repulsive bias (panel C). (**B**) Serial dependence on previous stimulus, considering trials for which the previous stimulus was presented at the same location as the current stimulus ('same location', purple), or trials for which the previous and current stimulus location was 10 visual degrees apart ('different location', orange). Attractive serial dependence is similarly strong for same and different location trials. Shaded region depicts the SEM of the group moving average (thin lines). Thick lines show the best fitting DoG curves. Same data as shown in Experiment 1 by *Fritsche et al., 2017*. (**C**) Average serial dependence on 4- to 9-back stimuli for same (purple) or different (orange) location trials. Current responses are more strongly repelled from 4- to 9-back stimuli, when current and past stimuli were presented at the same spatial location.

The online version of this article includes the following source data for figure 3:

**Source data 1.** Results of Experiment 2: Long-term repulsive biases are spatially specific.

repulsion bias for stimuli presented at a different spatial location is consistent with previous findings, which suggested that weak tilt-aftereffects may occur at spatial locations further away from the adaptor stimulus (e.g. see *Knapen et al., 2010*; *Mathôt and Theeuwes, 2013*).

## Experiment 3: Long-term repulsive biases are not strongly modulated by working memory delay

We have previously shown that the short-term positive bias exerted by the previous stimulus grows during the working memory retention period of the current trial, indicating a role of working memory for the positive bias (Experiment 4 in *Fritsche et al., 2017* and *Figure 4B*; see also *Bliss et al., 2017*; *Papadimitriou et al., 2015*). One may surmise that if the long-term repulsive bias reflects a low-level encoding bias, affecting the earliest stages of visual processing, then in contrast to the short-term positive bias, long-term repulsion should not be modulated by the duration of the current working memory delay. To test this hypothesis, we analyzed an additional, previously published dataset involving perceptual estimation responses to oriented Gabor stimuli, in which we introduced a variable delay period of 50 or 3500 ms between stimulus offset and the estimation response (*Fritsche et al., 2017*). Similar to Experiment 1 and 2, we found that estimation responses were biased towards the stimulus orientation of the previous trial (*Figure 4A*, max. attraction = 1.25°, p<0.0001), but repelled from stimuli presented further in the past (max. repulsion of −0.81° for 3-back stimulus, p<0.0001). This further corroborates the existence of short-term attraction and long-term repulsion biases in perceptual estimates. Interestingly, in comparison to Experiment 1 and 2, it appeared that the transition from attractive to repulsive effects occurred earlier in time, such that repulsive effects were already visible for the 2-back trial (*Figure 4A*, max. repulsion = −0.46°, p=0.034). Importantly, trials in Experiment 3 were longer compared to the previous experiments, due to the working-memory delay period (8.25 s vs. 5.37 and 6.55 s trial lengths). Thus, one potential explanation for the earlier onset of repulsion biases could be that the decay of attractive and repulsive biases may depend on the elapsed time (i.e. seconds) between previous and current

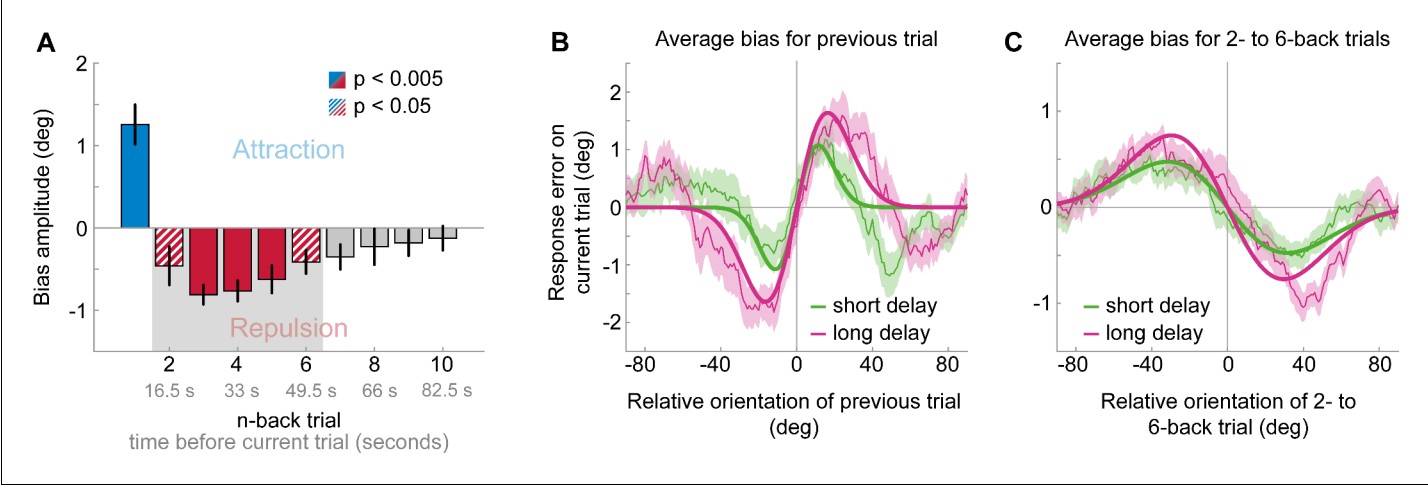

**Figure 4.** Results of Experiment 3: Long-term repulsive biases are not strongly modulated by working memory delay. (A) Attraction and repulsion biases exerted by the 10 preceding stimuli, pooled across response delay conditions. The current response is attracted towards the previous stimulus, but repelled from stimuli seen 2 to 6 trials ago. Colored bars indicate significant attraction (blue) and repulsion (red) biases (solid: Bonferroni correction for multiple comparisons; striped: no multiple comparison correction). Error bars represent 1 SD of the bootstrap distribution. The gray shaded area marks the time window of interest for which we estimated the working memory delay dependence of the repulsive bias (panel C). (B) Serial dependence on previous stimulus, considering trials for which the delay between current stimulus and response was short (green) or long (pink). Attractive serial dependence increases with working memory delay on the current trial. Shaded region depicts the SEM of the group moving average (thin lines). Thick lines show the best fitting DoG curves. Same data shown in Experiment 4 by *Fritsche et al., 2017*. (C) Average serial dependence on 2- to 6-back stimuli for current short (green) or long (pink) working memory delays. There is a trend towards a stronger repulsive bias when the current working memory delay is long. However, this difference between delay conditions was not significant.

The online version of this article includes the following source data for figure 4:

**Source data 1.** Results of Experiment 3: Long-term repulsive biases are not strongly modulated by working memory delay.

stimulus, rather than the number of intervening stimuli or decisions. We will return to a quantitative comparison of the temporal decay across the different experiments later.

The attraction bias towards the 1-back stimulus was stronger when the current memory delay was long compared to short (*Figure 4B*; short delay: max. attraction = 1.08°, p=0.001; long delay: max. attraction = 1.64°, p<0.0001; short vs. long: p=0.047; *Fritsche et al., 2017*). In contrast, for stimuli that exhibited overall repulsive biases (2- to six trials back, 17 to 50 s in the past), there was no significant difference in the magnitude of repulsion across memory delays (*Figure 4C*; short delay: max. repulsion = −0.47°, p=0.0002; long delay: max. repulsion = −0.75°, p<0.0001; short vs. long: p=0.055), although there was a trend towards stronger repulsion for current long delays. Indeed, there was a significant interaction between n-back trial (1-back vs. 2- to 6-back) and working memory delay ($F_{(1,23)}$ = 5.21, p=0.032, $\eta^2$ = 0.19), indicating that working memory delay has a stronger impact on short-term attraction than on long-term repulsion biases.

In order to quantify evidence for the null hypothesis of no modulation of the repulsive bias by working memory delay, we turned to a model-free analysis of averaging response errors for trials that were preceded by clockwise or counter-clockwise stimulus orientations, respectively (see Materials and methods). This allowed us to quantify serial dependence biases at the single participant level, which could subsequently be used for a Bayes Factor analysis. While this analysis provided moderate evidence for a modulation of the 1-back attraction bias by working memory delay ($BF_{10}$ = 7.25), which is in line with the model-based analysis, there was anecdotal (but no decisive) evidence for a lack of modulation for the long-term repulsive bias ($BF_{10}$ = 0.77).

## Experiment 4: Long-term history alters perceived orientation

To further test whether long-term repulsive biases arise at an early encoding stage, we turned to a different experimental paradigm, which sought to measure perception more directly compared to the stimulus reproduction technique employed in the previous experiments. Stimulus reproductions

necessarily rely on memory of the perceived stimulus feature and thus may reflect a combination of early perceptual and late post-perceptual working memory and decision biases (*Bliss et al., 2017*; *Fritsche et al., 2017*; *Papadimitriou et al., 2015*). We have previously shown that when the perceived orientation of a stimulus is measured more directly, by an immediate comparison to a reference stimulus that is visible at the same time (*Figure 5A*), it is repelled from the previous stimulus in a spatially specific manner, likely reflecting sensory adaptation (see *Fritsche et al., 2017* and *Figure 5C* 'current inducer'). Here we asked, whether the perceived orientation of the current stimulus, as measured in direct comparison to a reference, would continue to be repelled from stimuli experienced further in the past. This analysis not only allowed us to investigate whether the long-term repulsive bias reflects a genuine low-level perceptual bias that directly alters the appearance of a stimulus, as would be expected of sensory adaptation, but also enabled us to assess the temporal decay of the repulsion effect while short-term attraction biases are absent or strongly reduced (*Fritsche et al., 2017*; but see *Cicchini et al., 2017*). In Experiment 4, the same 24 observers that participated in Experiment 2 performed two consecutive tasks on each trial. First, they were simultaneously presented with two Gabor stimuli and were cued to reproduce the orientation of one of the stimuli by adjusting a response bar. We termed the cued stimulus 'inducer', since it induced a perceptual decision at one location. In order to measure subsequent biases in perception, another two Gabor stimuli were presented (one at the previously cued location and one at a location in the opposite hemifield, 20 visual degrees apart) and observers judged which of the two stimuli was tilted more clockwise (two-alternative forced choice [2AFC], *Figure 5A and B*). To recapitulate, we previously found in the same dataset that in this direct perceptual comparison the perceived orientations of the 2AFC stimuli were repelled from the immediately preceding inducer in a spatially specific manner (see *Fritsche et al., 2017* and *Figure 5C* 'current inducer'; same location: repulsion $-0.71°\pm$ 0.10 SEM, t(23) = $-6.91$, p=5e-7, d = $-1.41$; different location: repulsion $-0.17°\pm0.11$ SEM, t(23) = $-1.54$, p=0.14, d = $-0.31$; same vs. different: t(23) = $-3.85$, p=0.0008, d = $-0.79$). Importantly, here we found that this repulsive bias was not limited to the inducer presented on the current trial (*Figure 5C*). On average, inducers presented on the 10 previous trials, corresponding to a time window of 10 to 100 s prior to the current trial, exerted a significantly repulsive bias on the perceived orientation of the current 2AFC stimuli, when presented at an overlapping spatial location (same location: repulsion $-0.24°\pm0.05$ SEM, t(23) = $-5.24$, p=3e-5, d = - 1.07), but not when presented at a different location 10 visual degrees away (different location: repulsion $-0.09°\pm0.05$ SEM, t(23) = $-1.66$, p=0.11, d = 0.34; same vs. different location: t(23) = $-2.38$, p=0.026, d = $-0.49$). The temporal decay of this spatially specific repulsion bias was captured by an exponential decay model, which revealed a half-life of about five trials, corresponding to 50 s. The presence of long-term repulsive biases in perceptual comparisons indicates that brief stimuli experienced tens of seconds ago can directly bias the perceived orientation of a current stimulus in a spatially specific manner. This further supports the view that the long-term repulsive biases occur early during perceptual processing during the encoding of sensory information. Moreover, in the absence of the attractive bias measured in perceptual estimates (Experiment 1–3), the repulsive adaptation effect is strongest for stimuli which are temporally closest to the current stimulus and decays with increasing time between previous and current stimulus, with a long half-life. The current finding suggests that attractive and repulsive biases may coexist in perceptual estimations (Experiment 1–3), but differ in their relative strength and rate of decay. While both attractive and repulsive biases are strongest for the most recent stimulus, an initially stronger attractive bias typically masks a weaker repulsive bias, leading to an overall attractive influence of the immediate past. However, since the attractive bias decays at a much faster rate compared to the repulsive bias, the net bias that a stimulus exerts on the current perceptual estimate switches from attractive to repulsive as the intervening time increases.

## Repulsive and attractive biases can be explained by efficient encoding and optimal decoding of visual information

What are the underlying computational principles that may lead to these concurrent attractive and repulsive biases in perceptual decisions? To shed more light on this question, we turned to computational modeling to explore whether the coexisting attractive and repulsive biases can be explained by an observer that exploits the temporal continuity of visual environment in order to optimize the efficient *encoding* and/or *decoding* of visual information.

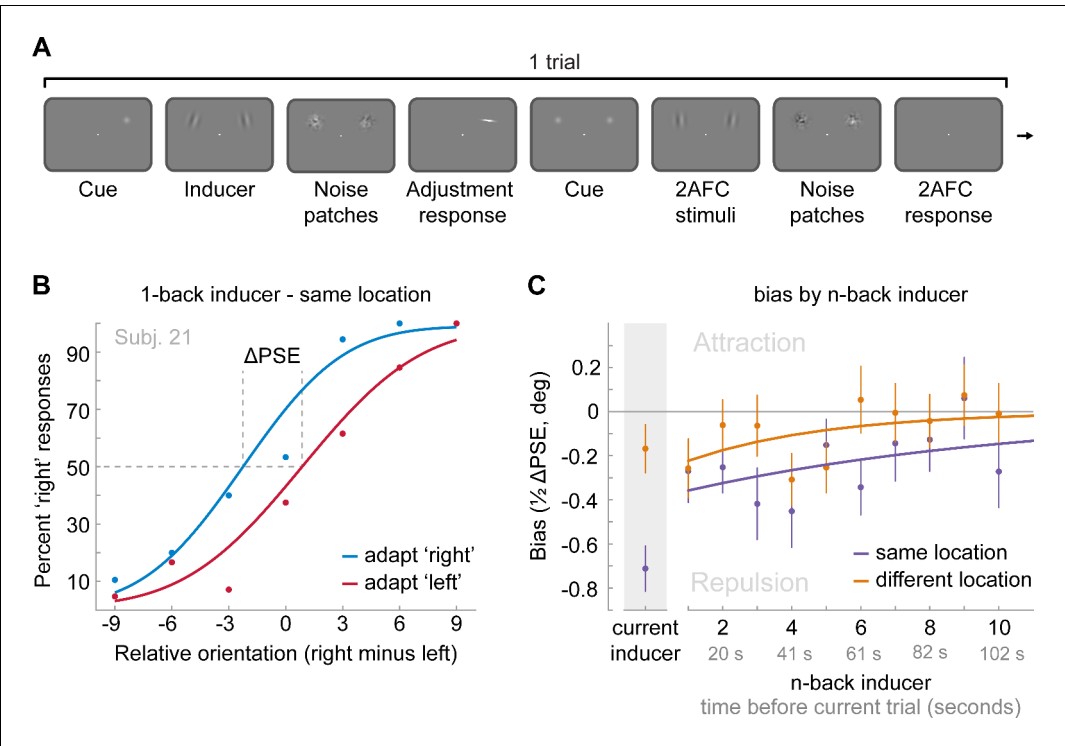

**Figure 5.** Task and results of Experiment 4: The long-term stimulus history directly biases the perceived orientation of current stimuli. (**A**) Observers were cued to reproduce one of two Gabor stimuli by adjusting a response bar (adjustment response). Subsequently, two new Gabor stimuli appeared at priorly cued locations in the left and right visual field. Those stimuli could appear either at the same locations as the previous stimuli or 10 visual degrees above or below. Observers had to judge which of the two new stimuli was oriented more clockwise (2AFC). Similar to the previous experiments, all Gabor stimuli were followed by noise masks. (**B**) Psychometric curves for the 2AFC judgments of one example observer. We expressed the probability of a 'right' response (y-axis) as a function of the orientation difference between right and left 2AFC stimuli (x-axis). For positive x-values, the right stimulus was oriented more clockwise. We binned the trials in two bins according to the expected influence of repulsive adaptation to the inducer stimulus. Blue data points represent trials in which repulsive adaptation, away from the inducer, would favor a 'right' response, while red data points represent trials in which it would favor a 'left' response. The example observer exhibits a repulsive adaptation bias. We quantified the magnitude of the bias as the difference in the points of subjective equality between 'adapt right' and 'adapt left' curves (ΔPSE) divided in half. This value indicates how much a 2AFC stimulus is biased by a single inducer stimulus. (**C**) Bias exerted by the current inducer (gray shaded box, same data as in Experiment 2 by *Fritsche et al., 2017*) and inducers of the 10 preceding trials. Previous inducer stimuli exert a repulsive bias on the current 2AFC stimuli, specifically when past inducer and current 2AFC stimuli were presented at an overlapping spatial location. This bias appears to decay exponentially for inducers encountered further in the past. Note that while current inducers were always oriented ±20° from the subsequent 2AFC stimulus, constituting an optimal orientation difference for eliciting adaptation biases (e.g. see *Figure 2B*), relative orientations of previous inducers ranged from −49 to 49°, and thus presented overall less effective adaptor stimuli. This may explain the discontinuity in the magnitude of repulsion exerted by the current and previous trial inducer. Data points show group averages and error bars present SEMs.

The online version of this article includes the following source data for figure 5:

**Source data 1.** Task and results of Experiment 4: The long-term stimulus history directly biases the perceived orientation of current stimuli.

---

Previous studies proposed that attractive biases in perceptual estimates arise from a probabilistically optimal strategy of decoding sensory information into a perceptual decision (*Cicchini et al., 2018*; *van Bergen and Jehee, 2019*). Attractive serial dependencies towards the previous stimulus orientation are well captured by such an observer model that estimates the current stimulus orientation by integrating a noisy sensory measurement of the current stimulus with a prior prediction about

the upcoming stimulus orientation in a probabilistically optimal, that is Bayesian, manner (*Figure 6A*, orange box). If predictions are well matched to the temporal regularities in the environment, combining predictions based on previous stimuli with current sensory measurements generally produces more accurate stimulus estimates than those based on current sensory measurements alone. We implemented such an ideal observer model to test whether it could account for the short-term attractive and long-term repulsive biases in the empirical data. In this observer model, the prediction about the upcoming visual stimulus was formed by combining an internal model of naturally occurring orientation changes (*Figure 6A*, green box, *transition distribution*) with knowledge about previous stimulus orientations (i.e. previous posterior distributions). In contrast to the ideal observer model by *van Bergen and Jehee, 2019*, we here assumed that the observer does not only rely on a prediction formed on the most recent trial, but utilizes a weighted mixture of previous predictions, where the relative contribution of previous predictions is determined by an exponential integration time constant (*Figure 6A*, blue box, *temporal integration weights*). Such a mixture prior accounts for the fact that statistical regularities in natural environments do not only exist between immediately consecutive orientation samples, but over extended timescales, with the most recent stimuli also being the most predictive of the current stimulus. In contrast to an observer that only relies on the prediction based on the previous trial, an observer utilizing such a mixture prior has been shown to adequately capture positive serial dependencies beyond the most recent trial (*Kalm and Norris, 2018*). We found that this observer model indeed captured attractive serial dependencies. However, it was not able to produce the long-term repulsive biases found in Experiments 1–3 (*Figure 7* and *Figure 7—figure supplement 2*). Thus, an observer that exploits the temporal stability of the environment via Bayes-optimal decoding alone cannot explain the pattern of attractive and repulsive biases. Furthermore, the inability of the Bayesian ideal observer to capture both attraction and repulsion biases provides evidence that the long-term repulsion biases are not an artifact resulting from short-term attraction biases and random fluctuations in the input sequence (*Maus et al., 2013*).

As described in the Introduction, repulsive adaptation biases have been proposed to arise from efficient encoding of visual information (*Barlow and Földiák, 1989*; *Clifford et al., 2000*; *Müller et al., 1999*; *Stocker and Simoncelli, 2006*). For instance, the sensory system may allocate sensory coding resources such that it is optimally adapted to represent the current stimulus (*Wei and Stocker, 2017*; *Wei and Stocker, 2015*). For temporally stable visual input, in which current stimulus features are similar to previous ones, an optimally efficient allocation of encoding resources entails that more resources should be dedicated to stimulus features encountered in the recent history. As a consequence, the most likely stimulus, as predicted from the sensory history, is represented with highest fidelity. Here, we implemented an observer model that predicted the orientation of an upcoming stimulus based on previous sensory measurements and subsequently allocated sensory encoding resources such that the mutual information between the predicted stimulus and its sensory representation was maximized (*Figure 6B*; *Wei and Stocker, 2015* and '*Efficient encoding model with history-independent decoding*' in Materials and methods). For this observer, repulsive adaptation biases then result not from changes to the prior, but from changes in the sensory representation of the current stimulus; in other words in the likelihood function. Due to biased allocation of encoding resources, the likelihood function of a current stimulus often becomes asymmetric, with a long tail away from the most likely predicted stimulus orientation (*Wei and Stocker, 2015*; *Figure 6B*). Any subsequent 'readouts' of this likelihood function by downstream decoders, which take the full likelihood function into account, will consequently be biased away from the predicted stimulus.

We found that such an observer with optimally efficient encoding at the likelihood stage, but no influence of the sensory history on the decoding stage (i.e. a uniform, history-independent decoding prior), is able to produce long-term repulsive biases. However, this observer additionally results in repulsive and not attractive biases in response to the most recently experienced stimuli (*Figure 7—figure supplement 2*). Therefore, an observer with efficient coding alone is not able to capture the transition from short-term attractive to long-term repulsive biases observed in the empirical data.

Neither an observer in which sensory history influenced only Bayesian decoding nor only efficient encoding was able to reproduce the pattern of attractive and repulsive biases that we observed in perceptual decisions. An intriguing possibility is that the coexisting attractive and repulsive biases might arise from a concurrent optimization of both encoding and decoding according to the sensory history. We tested this idea by developing an ideal observer model with both efficient encoding and

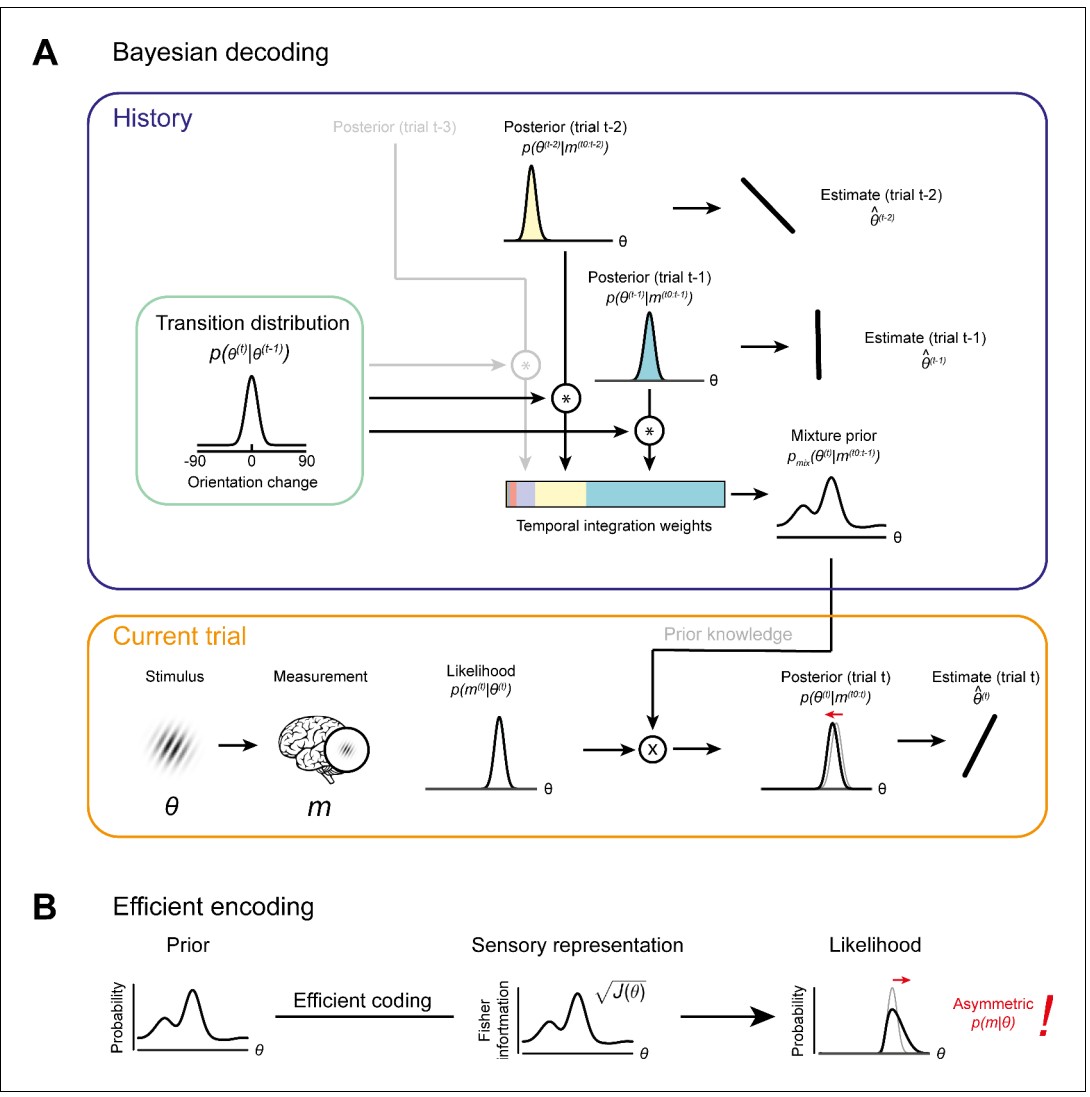

**Figure 6.** Bayesian decoding and efficient encoding of orientation information in a stable environment. (**A**) Bayesian decoding. Orange box: The observer encodes a grating stimulus with orientation $\theta$ into a noisy measurement $m$. Since the noisy measurement is uncertain, it is consistent with a range of orientations, described by the likelihood function. The likelihood is combined with prior knowledge to form a posterior, which describes the observer's knowledge about the current stimulus orientation. The final orientation estimate is taken as the posterior mean. Blue box: In a stable environment, the observer can leverage knowledge about previous stimuli for improving the current estimate. To predict the current stimulus orientation, the observer combines a model of orientation changes in a stable environment, represented by a transition distribution (green box), with knowledge about previous stimuli, that is previous posteriors. Predictions based on previous stimuli are integrated into recency-weighted mixture prior, using exponential integration weights. This mixture prior is subsequently used for Bayesian inference about the current stimulus. (**B**) Efficient encoding. The observer maximizes the mutual information between the sensory representation and physical stimulus orientations by matching the encoding accuracy (measured as the square root of Fisher information $J(\theta)$) to the prior probability distribution over current stimulus orientations. In a stable environment, this prior distribution can be informed by previous sensory measurements. With some assumptions about the sensory noise characteristics (see Materials and methods), the likelihood function of new sensory measurements is fully constrained by the Fisher information. The likelihood function is typically asymmetric, with a long tail away from the most likely predicted stimulus orientation. For details see Materials and methods and *Wei and Stocker, 2015*.

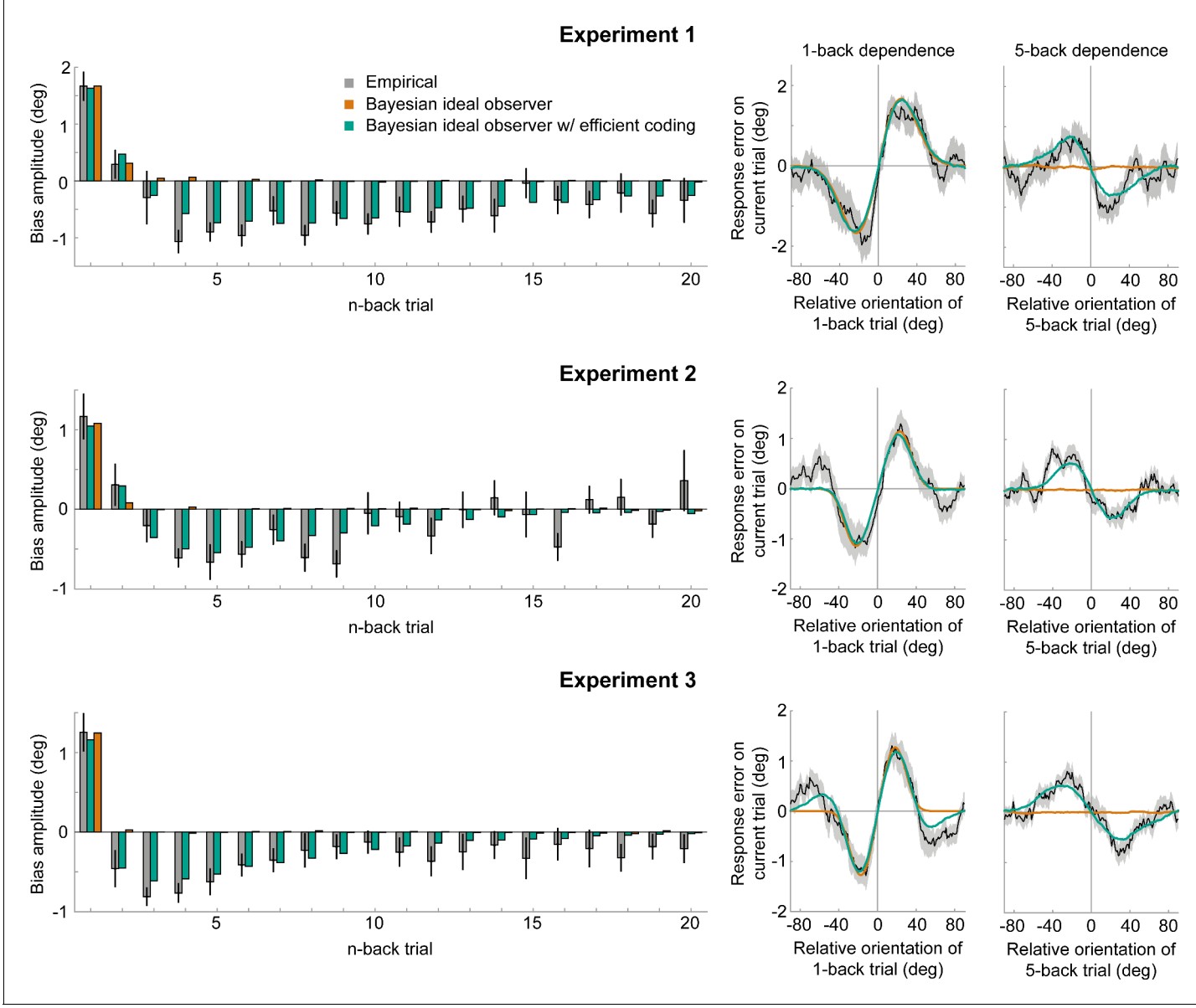

**Figure 7.** Empirical biases and ideal observer predictions. Left column: An observer with efficient encoding and history-dependent Bayesian decoding (green) accurately captures the empirical magnitudes of short-term attractive and long-term repulsive biases exerted by the 1- to 20-back trials of Experiment 1–3 (grey). In contrast, an observer with Bayesian decoding (orange) only produces positive biases, but cannot capture the long-term repulsive biases. For each experiment, model bias amplitudes were estimated by simulating the observer model with the best fitting set of parameters on the stimulus sequences shown to human participants, and fitting the resulting model response errors with a Derivative-of-Gaussian (DoG) curve. Analogous to the analysis of human behavior, the amplitude parameter of the DoG curve was taken as the bias amplitude. Error bars represent 1 SD of the bootstrap distribution of the empirical data. Right column: Both attractive (left) and repulsive biases (right) of the ideal observer model with efficient encoding and Bayesian decoding (green) closely follow the average biases of human participants (black). The observer with Bayesian decoding (orange) does not produce repulsive biases. Black shaded regions show the SEMs of the empirical data. Model fits to the full range of 1- to 20-back conditioned response errors are shown in *Figure 7—figure supplements 1* and *2*. A comparison of cross-validated prediction accuracies of the different models is shown in *Figure 7—figure supplement 3*. The best fitting parameters of the observer model with efficient encoding and history-dependent Bayesian decoding are reported in *Figure 7—figure supplement 5*.

The online version of this article includes the following source data and figure supplement(s) for figure 7:

**Source data 1.** Empirical biases and ideal observer predictions.

**Figure supplement 1.** Serial dependence biases of an ideal observer with efficient encoding and history-dependent Bayesian decoding (green).

**Figure supplement 2.** Serial dependence biases of an ideal observer, in which sensory history influenced only Bayesian decoding (orange) or only efficient encoding (blue).

**Figure supplement 3.** Cross-validated prediction accuracies of the four different observer models.

*Figure 7 continued on next page*

*Figure 7 continued*

**Figure supplement 4.** Normalized variability of estimation response errors as a function of the orientation difference between current and n-back trial.
**Figure supplement 5.** Best fitting parameters of the observer model with efficient encoding and history-dependent Bayesian decoding (Full efficient-encoding-Bayesian-decoding model).

history-dependent Bayes optimal decoding of sensory information (see '*Full efficient-encoding-Bayesian-decoding model*' in Materials and methods). In this observer, efficient coding determined the shape of the likelihood function during the encoding of the current stimulus orientation, whereas Bayesian decoding prescribed how this likelihood function was subsequently integrated with a prior prediction about the current stimulus orientation. Importantly, we allowed predictions used for optimizing encoding and decoding to be based on potentially distinct transition distributions and integration time constants. This was motivated by the possibility that encoding transition models in low-level sensory circuits could be relatively inflexible and learned over the organism's lifetime, whereas higher-level transition models used for decoding are likely context- and task-dependent (*Fischer et al., 2020*) and learned quickly, within minutes to hours (*Braun et al., 2018*).

We found that this ideal observer model with efficient encoding and history-dependent Bayesian decoding was able to faithfully reproduce short-term attraction and long-term repulsion biases across all three experiments (*Figure 7* and *Figure 7—figre supplement 1*). It predicted estimation biases better than observers, in which sensory history only influenced Bayesian decoding or only efficient encoding (*Figure 7—figre supplement 3*; cross-validated prediction accuracy, Exp. 1: $r = 0.58$ vs. 0.29 and 0.27; Exp. 2: $r = 0.53$ vs. 0.38 and 0.20; Exp. 3: $r = 0.66$ vs. 0.32 and 0.42). Furthermore, the observer captured estimation biases far better than an observer in which efficient encoding and Bayesian decoding were based on the same predictions, using the identical transition distribution and integration time constant (see "*Efficient-encoding-Bayesian-decoding model (single prior)*" in Materials and methods; cross-validated prediction accuracy, Exp. 1: $r = 0.58$ vs. 0.29; Exp. 2: $r = 0.53$ vs. 0.20; Exp. 3: $r = 0.66$ vs. 0.42). Crucially, we found that predictions for efficient encoding and Bayesian decoding were integrated over different timescales. Bayesian decoding was based on predictions with a short half-life, thus only integrating information of the most recent trials (decoding $t_{1/2}$: Exp. 1: 4.59 s; Exp. 2: 4.47 s; Exp. 3: 3.94 s). Conversely, predictions used for efficient encoding were integrated over a longer timescale, as indicated by substantially longer half-lives (encoding $t_{1/2}$: Exp. 1: 40.10 s; Exp. 2: 18.94 s; Exp. 3: 30.99 s). Moreover, we observed that transition distributions, reflecting internal models of orientation changes occurring in the environment, were broader at the encoding compared to decoding stage across all three experiments (Exp. 1: $\sigma_{tr} = 11.28°$ vs 8.35°; Exp. 2: $\sigma_{tr} = 13.02°$ vs 9.14°; Exp. 3: $\sigma_{tr} = 21.77°$ vs 9.75°). This suggests that the observer expects orientation information to be more variable at encoding compared to decoding. One speculative explanation is that the encoding transition model captures more variable orientation changes in spatially localized, retinotopic regions of the visual field, whereas the decoding transition model captures more stable orientation transitions between successively attended features across a broad region of visual space.

Finally, we sought to corroborate evidence for the involvement of efficient encoding principles in the present experiments. The current efficient coding scheme prescribes that more encoding resources should be dedicated to stimulus features encountered in the recent history. Next to inducing repulsive biases, such a reallocation of encoding resources would increase the fidelity of stimulus representations that are similar to recently encountered orientations, by narrowing the width of the likelihood function. From this follows the prediction that estimation responses should be less variable when current and previous stimulus orientation are similar. We indeed observed a marked reduction in response variability when the current and 1-back stimulus were of similar orientation (*SD* reduction – Exp. 1: −0.99°±0.18; Exp. 2: −0.59°±0.14; Exp. 3: −0.62°±0.12; *Figure 7—figure supplement 4*; for details see Materials and methods). However, variability estimates for temporally more distant stimuli were noisy and did not exhibit a consistent reduction for similarly oriented n-back stimulus orientations (*Figure 7—figure supplement 4*), so we did not find evidence for a slow decay in variability, as we reported for the repulsive bias.

Together, the current findings indicate that the coexisting attractive and repulsive biases in perceptual decisions may originate from the concurrent optimization of efficient encoding and Bayes-

optimal decoding of visual information. Furthermore, our model suggests that human observers use distinct predictions, accumulated over different timescales of the perceptual history, to optimize encoding and decoding of sensory information in a stable environment.

## Discussion

The goal of the current study was to disentangle and understand repulsive and attractive biases in perceptual decision making by exploring the respective timescales over which current visual processing is influenced by the perceptual history. Across four experiments, we show that perceptual estimates are attracted towards recently seen stimuli, but repelled from stimuli encountered further in the past. The long-term repulsive bias was spatially specific, directly altered the perceived orientation of the current stimulus, and appeared to be only weakly or not at all modulated by the current working memory delay. These findings clearly point towards a low-level sensory encoding origin of the long-term repulsive bias. In contrast, attraction biases were short-lived, spatially unspecific and they were modulated by the working memory delay on the current trial. To further disentangle attractive and repulsive biases and to elucidate the computational principles underlying these history dependencies, we developed an ideal observer model that combined efficient encoding with Bayes-optimal decoding of sensory information. This model was able to capture both short-term attractive and long-term repulsive serial dependencies across all perceptual estimation experiments. Exploiting the temporal stability of the environment according to efficient encoding and Bayesian decoding principles may enable our brains to achieve two concurrent goals of perceptual decision-making. By efficiently allocating sensory encoding resources, the visual system can maximize its sensitivity to small changes in the environment. Concurrently, Bayes-optimal integration of current noisy sensory measurements with predictions based on recent sensory input guards visual processing against instabilities, effectively smoothing visual representations across time. This ideal observer model thus illustrates a principled way in which observers can exploit the temporal stability in the environment, using different mechanisms across multiple processing stages, to optimize visual processing.

### Distinct timescales of attractive and repulsive serial dependencies

Across three perceptual estimation experiments, we found that while the current estimation response is attracted towards the most recently perceived stimuli, it is simultaneously repelled from stimuli seen further in the past. Based on our ideal observer model, we estimate that the attractive influence of a previous stimulus decreases exponentially and is reduced to half of its initial magnitude after ~4.5 s. This exponential half-life is in line with previous studies, which found attractive serial dependencies to decay over a relatively short time period of about 10–15 s (*Fischer and Whitney, 2014*; *Suárez-Pinilla et al., 2018*; *Van der Burg et al., 2019*). Conversely, timescale estimates of repulsive adaptation effects in response to brief stimuli have been more variable. Several studies have reported short-lived adaptation, persisting no more than a few seconds (*Kanai and Verstraten, 2005*; *Pastukhov and Braun, 2013*; *Pastukhov and Braun, 2013*; *Pavan et al., 2012*). However, recent studies found that initial attraction biases are replaced by long-term repulsive biases that extend many trials into the past (*Gekas et al., 2019*; *Gordon et al., 2019*; *Suárez-Pinilla et al., 2018*), potentially lasting for minutes (*Chopin and Mamassian, 2012*; but see *Maus et al., 2013*). Our current findings are in line with these latter reports of long-term repulsive adaptation. We estimate the repulsive bias reaches half of its initial magnitude after 20 to 40 s and this is thus marked by a distinctly slower decay compared to the short-term attraction bias. Importantly, several of the previous studies reporting long-term repulsive biases measured perceptual decisions in two-alternative forced choice paradigms (*Gordon et al., 2019*), while employing repeated presentations of a limited set of stimuli (*Chopin and Mamassian, 2012*; *Gekas et al., 2019*). In the current study we demonstrate that long-term repulsive biases are similarly present in two-alternative forced choice (Experiment 4) *and* continuous report paradigms (Experiment 1–3), even when presenting continuously varying stimuli from the full range of possible orientations. Thus, repulsive biases are unlikely to arise due to artifacts of the measurement method or necessitate the repeated presentation of the same stimuli, but present a genuine perceptual bias arising from the brief exposure to individual stimuli experienced seconds to minutes in the past.

Notably, opposite effects to ours have also been reported in the literature (*Dekel and Sagi, 2015*). In particular, Dekel and Sagi observed short-term repulsive biases followed long-term

attraction, when observers adapted to natural image stimuli. However, note that the long-term attraction biases were likely explained by response correlations, which are absent in the present experiments (see *Dekel and Sagi, 2015*, sections 3.4 and 4.4.).

One outstanding question is whether short-term attractive and long-term repulsive biases decay as a function of physical time, or as a function of intervening visual events. We estimated relatively similar half-lives of the temporal decay of both attractive and repulsive biases across estimation experiments, even though experiments differed in their average trial durations (5.37, 6.55 and 8.25 s). This may suggest the decay is mainly determined by intervening time and not the frequency of stimulus presentations. However, estimation experiments also differed in other aspects of experimental design, complicating a direct comparison between experiments. Furthermore, the only source of temporal variability within the current experiments was introduced via the self-paced estimation responses, leading to only minor temporal jitter across trials (in other words, intervening time and intervening trials were very highly correlated). This renders the current experiments unsuitable for distinguishing the role of elapsed time and intervening stimulus presentations for the history decay. Future studies may investigate this question more specifically, by experimentally dissociating time and the frequency of stimulus presentations.

## Repulsive and attractive biases can be explained by efficient encoding and Bayesian decoding of visual information

Previous studies proposed that attractive biases in perceptual estimates towards the previous stimulus arise from a probabilistically optimal strategy of decoding sensory information into a perceptual decision (*Cicchini et al., 2018*; *Kalm and Norris, 2018*; *van Bergen and Jehee, 2019*). In a stable environment, integrating current noisy sensory measurements with information about previous stimuli generally results in more accurate perceptual estimates, compared to relying on noisy sensory measurements alone. Effectively, such a Bayesian integration of previous and current visual information would smooth perceptual representations over time, thereby promoting visual stability in the face of disruptive factors, such as eye blinks and external and internal noise. However, in the current study, we show that the temporal pattern of history biases is more complex, consisting of a superposition of concurrent attractive and repulsive biases, which cannot be explained by an observer model based on Bayesian decoding of visual information alone. Importantly, this not only shows that a Bayesian observer model is inadequate to explain the full pattern of history dependencies in perceptual estimates, but the inability of the Bayesian observer to capture both short-term attraction and long-term repulsion biases also indicates that the long-term repulsion biases are not an artifact resulting from short-term attraction biases in combination with random input fluctuations (*Maus et al., 2013*). Rather, the spatial specificity (Experiment 2) and the presence in immediate perceptual comparisons to a reference stimulus (Experiment 4) suggest a low-level sensory origin of the long-term repulsive bias, akin to sensory adaptation (*Gibson and Radner, 1937*).

Prominent theories of sensory adaptation posit that repulsive biases may arise as a consequence of optimally efficient encoding of visual information in a stable environment. In particular, repulsive biases may arise due to self-calibration and decorrelation (*Barlow and Földiák, 1989*; *Clifford et al., 2000*; *Müller et al., 1999*), or an increase in the signal-to-noise ratio for measurements similar to recently perceived stimuli (*Stocker and Simoncelli, 2006*). In line with the latter proposal, we here developed a model in which repulsive adaptation biases arise as a consequence of efficient allocation of sensory coding resources (*Wei and Stocker, 2017*; *Wei and Stocker, 2015*). Since in a temporally stable visual environment new visual input is likely to be similar to recent input, the visual system can exploit the sensory history to strategically allocate sensory coding resources such that the mutual information between the likely physical stimulus and its sensory representation is maximized. Such an efficient allocation of sensory resources increases sensitivity to small changes in the input, in line with previous findings of improved discriminability following adaptation (*Clifford et al., 2001*; *Mattar et al., 2018*; *Regan and Beverley, 1985*). At the same time, it leads to a repulsive bias of perceptual estimates, away from previous stimulus orientations, thus producing classical sensory adaptation effects.

Across three experiments involving perceptual estimation tasks, we show that an ideal observer with efficient encoding together with Bayesian decoding can adequately capture the pattern of short-term attraction and long-term repulsion biases in perceptual estimates. This observer model provides a framework for explaining how the visual system exploits temporal regularities in the input

to achieve sensitivity to changes in the environment, while at the same time maintaining stable visual processing of visual information. Importantly, however, our model indicates that efficient encoding and Bayesian decoding are based on distinct priors, accumulating previous stimulus information over different timescales and employing distinct internal models of naturally occurring input changes. In particular, at the encoding stage, the prediction about the upcoming stimulus is based on many previous stimulus encodings, reaching far into the past. Conversely, the prediction used for Bayesian decoding is solely based on previous stimulus information encountered in the most recent past (previous 2–3 trials). Moreover, the transition distribution, reflecting an internal model of naturally occurring orientation changes, is broader at the encoding compared to the decoding stage, suggesting that the observer expects successive orientation samples to be less stable at encoding compared to decoding. As a consequence, the efficient allocation of encoding resources is less biased to any particular previous stimulus, but more broadly influenced by the long-term history of visual stimulation. In contrast, at decoding, the relatively stronger contribution of the short-term history to the Bayesian decoding prior leads to a dominating attractive influence of the most recent stimuli, but no influence of long-term stimulus history.

We may speculate on the reasons for the distinct timescales and transition models for efficient encoding and Bayesian decoding. At encoding, the long integration timescale and broad transition model, which effectively diffuse and smooth prior information across time, may reflect the need to adapt to long-term perturbations in visual processing or slowly changing input statistics without over-adjusting to the rapidly changing low-level input at spatially localized, retinotopic regions of the visual field. In contrast, the more narrow transition model and short integration timescale at decoding may be tuned to more stable orientation transitions between briefly attended features across a broad region of visual space. However, such explanations remain speculative at this moment. More extensive characterizations of the spatiotemporal statistics of natural visual input and visual information processing will be needed to advance our understanding of the principles underlying the distinct encoding and decoding priors. In this regard, it will also be interesting to study whether observers can flexibly modify encoding and/or decoding transition models in response to experimentally induced temporal regularities in visual input. There is evidence that human observers can indeed adapt to simple temporal regularities after relatively brief exposure (*Braun et al., 2018*), and our ideal observer model provides a framework to understand changes in internal transition models across multiple stages of the perceptual decision-making process.

The current model in which attraction biases are due to Bayesian decoding of sensory information provides a parsimonious explanation for the increase of attractive serial dependence biases with working memory delay (Experiment 3; *Bliss et al., 2017*; *Papadimitriou et al., 2015*), and the absence of short-term attraction biases in the perceptual comparison task of Experiment 4. In particular, Bayesian decoding posits that the attractive serial dependence bias is dependent on the relative widths of sensory likelihood and prior. When sensory likelihoods are relatively narrow, briefly after the initial encoding of the sensory stimulus, the prior has little influence on the resulting posterior, leading only to small attraction biases. As the sensory representation degrades during working memory retention, and the likelihood becomes increasingly broader, the influence of the prior increases. As a result, the posterior will be increasingly pulled towards the prior, leading to an increase in attraction during working memory retention. Importantly, in Experiment 4, observers could make their perceptual decision about the perceptual comparison stimuli while the stimuli were simultaneously presented on the screen, minimizing the influence of working memory noise and presumably leading to relatively narrow sensory likelihoods at the time of the perceptual decision. As outlined above, this would lead to a strong reduction or even absence of attraction biases in this scenario, consistent with the absence of attraction biases in Experiment 4. Furthermore, in agreement with this proposal, it has recently been found that attractive biases can arise early after stimulus offset when visual stimulation is very weak (*Manassi et al., 2018*), leading to a broad likelihood function, and are stronger for more uncertain stimulus representations (*Cicchini et al., 2018*; *van Bergen and Jehee, 2019*). Whether such a Bayesian decoding computation operates continuously during working memory retention, causing systematic drifts in working memory representations over time (*Panichello et al., 2019*; *Wolff et al., 2020*), or occurs only once during the final read-out of the working memory representation is an exciting question for future research.

The current model bears some similarity to a recently proposed two-process model of serial dependence in which repulsive biases are mediated by sensory adaptation, in the form of gain

changes in low-level perceptual neural populations, whereas attractive biases arise due to persistent read-out biases at a higher-level decisional units (*Pascucci et al., 2019*). A key distinction to the model by Pascucci et al. is that the current model of repulsive adaptation and attractive serial dependence is based on a normative framework of efficient encoding and Bayesian decoding of visual information. At the same time, the current model is agnostic to the neural implementation underlying efficient encoding and Bayesian decoding. In fact, at the level of neural population tuning, efficient encoding could be realized by multiple alternative mechanisms, involving changes in the location, width and/or gain of neural tuning curves of sensory neurons (*Wei and Stocker, 2015*). Similarly, Bayesian decoding could be achieved by combining predictions with sensory information in low-level sensory circuits via feedback mechanisms or later integration in mid- and high-level visual areas. The model by Pascucci et al., which models a potential neural implementation of history biases, and the current model are therefore not mutually exclusive, but rather reside at two different levels of explanation. Notably, however, if human visual processing indeed follows the computational goals of efficient encoding and Bayesian decoding of visual information, this will put important constraints on the neural mechanisms that need to fulfill these goals.

The general premise that repulsive biases arise at a low-level encoding stage, whereas attractive biases arise at a higher-level decoding stage is well supported by recent empirical findings. For instance, it has been shown that in contrast to repulsive adaptation, attractive serial dependence biases are strongly dependent on attention (*Fischer and Whitney, 2014*; *Fritsche and de Lange, 2019*; *Suárez-Pinilla et al., 2018*), and are modulated by subjective confidence (*Samaha et al., 2019*; *Suárez-Pinilla et al., 2018*). Furthermore, in contrast to repulsive adaptation biases, which exhibit a clear spatial specificity (*Figures 3* and *5*; *Knapen et al., 2010*), attractive serial dependencies appear to have a very broad spatial tuning (*Collins, 2019*; *Fischer and Whitney, 2014*; *Fritsche et al., 2017*) and have been shown do increase during the post-perceptual working-memory delay period (*Bliss et al., 2017*; *Fritsche et al., 2017*; *Papadimitriou et al., 2015*). A recent study showed that attractive serial dependencies are abolished when backward-masking the previous inducer stimulus, suggesting a role of high-level feedback processing in attractive serial-dependence (*Fornaciai and Park, 2019*). Conversely, repulsive adaptation was not affected by masking, consistent with feedforward changes in local, low-level neural circuits underlying adaptation. Moreover, a recent study found that attractive serial dependences can be modulated by contextual features, such as stimulus color or serial position in a trial, suggesting a more selective integration of prior information at the higher-level decoding stage (*Fischer et al., 2020*). Finally, there is evidence that repulsive adaptation and attractive serial dependence map onto distinct cortical networks (*Schwiedrzik et al., 2014*). While adaptation was confined to posterior regions, comprising early visual cortex, a more widespread network of higher-level visual and fronto-parietal areas was involved in attractive serial dependence.

Despite the above empirical evidence that points towards low- and high-level visual processing as the source of repulsive and attractive history biases, one may wonder to which extent the current biases could be explained by temporal dependencies in low-level motor processing instead. There are several arguments against a systematic influence of motor biases in the present experiments. First, in each trial of the perceptual estimation tasks (Experiments 1–3), the initial orientation of the response bar was randomly selected from the range of all possible orientations (0180], and thus the motor actions required for adjusting the response bar to a particular orientation were independent across trials. Second, several previous studies have shown that an explicit reproduction response is not necessary to elicit an attraction bias on the next trial (*Czoschke et al., 2019*; *Fischer and Whitney, 2014*; *Suárez-Pinilla et al., 2018*), that perceptual decisions are attracted towards the presented stimulus orientation rather than a mirrored response orientation (*Cicchini et al., 2017*), and that attraction biases occur even when performing a different task (and response) on the previous stimulus (*Fritsche and de Lange, 2019*). Together, these studies strongly suggest that short-term attractive serial dependencies are not due to low-level motor biases. Furthermore, it seems unlikely that low-level motor biases would lead to larger attraction biases with increasing working memory delay (Experiment 3) and that they would induce *spatially specific* long-term repulsion biases (Experiments 2 and 4). Additional evidence against an involvement of motor biases in the long-term repulsion effect is provided by Experiment 4. In this experiment, participants alternate between a reproduction task and 2-AFC task, which require very different motor responses. Furthermore, stimuli in Experiment 4 were counterbalanced such that simple motor biases in the 2-AFC task (e.g. a

previous response to a leftward tilted stimulus in the reproduction task induces a bias to press the left response button in the 2-AFC task) would not induce systematic biases in the analysis of the perceptual comparison data. Taken together, while there is evidence for temporal dependencies in low-level motor responses (*Pape and Siegel, 2016*), the above points provide strong arguments against a systematic influence of such dependencies on the current estimate of short-term attraction and long-term repulsion biases in perceptual decisions.

Finally, next to short-term attraction between successive stimuli with similar orientations, in Experiment 2 and 3 we observed systematic short-term repulsion when the orientation difference between current and previous stimuli was large (>60°; see *Figures 3B* and *4B*), in line with previous findings (*Fritsche and de Lange, 2019*; *van Bergen and Jehee, 2019*). In the current study, we specifically focused on short-term attraction biases between similar stimuli and the current ideal observer model does not capture the repulsion biases between highly dissimilar stimuli. We previously found that while the attraction bias between stimuli with similar orientations were strongly dependent on attention to previous stimulus features, the repulsion bias for dissimilar stimuli was not modulated by attention (*Fritsche and de Lange, 2019*). This dissociation speaks against a model in which attraction of similar and repulsion of dissimilar stimuli are mediated by the same Bayesian decoding process. Rather, the short-term repulsion effect could reflect an encoding bias, that is sensory adaptation. While the repulsion bias of the previous stimulus is indeed partly captured by the efficient encoding component in Experiment 3 (*Figure 7*, lower row), it is not accounted for in Experiment 2 (*Figure 7*, middle row). Furthermore, in Experiment 1, repulsion by previous dissimilar stimuli is absent, even though long-term repulsion effects are clearly present (*Figure 7*, upper row). This suggests that long-term and short-term repulsion biases are (at least partially) independent. Taken together, more research is necessary to elucidate the nature of the repulsion biases for large orientation differences and their potential relationship to efficient encoding and Bayesian decoding of sensory information.

## Limitations

It should be noted that current ideal observer model with efficient encoding and Bayesian decoding is based on several assumptions. For instance, we assumed a particular efficient coding scheme in which the observer maximizes the mutual information between physical stimuli and sensory representations (*Wei and Stocker, 2015*). Due to the efficient reallocation of encoding resources to stimulus features encountered in the recent history, this efficient encoding scheme makes the prediction that estimation responses should be less variable when current and previous stimulus orientation are similar. While we indeed observed a short-term reduction of response variability when current and previous stimuli were similar, variability estimates for temporally more distant stimuli were noisy and did not exhibit a consistent reduction for similarly oriented n-back stimulus orientations, so we did not find evidence for a slow decay in variability, as we reported for the repulsive bias. Thus, we were not able to fully confirm the predicted changes in encoding fidelity for temporally more distant trials. Future studies may specifically test whether long-term repulsive biases are accompanied by reductions in discrimination thresholds, as predicted from the efficient coding scheme based on the maximization of mutual information. There exist alternative formulations of efficient coding, such as minimizing redundancy (*Barlow, 1961*) or reconstruction error (*Wang et al., 2012*). Encoding efficiency may also depend on the specific task context for which the encoded information is used. Since low-level orientation information is likely used for many different tasks, the current efficient coding scheme in which a more generic information criterion is optimized may represent a good general strategy for the visual system (*Wei and Stocker, 2015*). A further assumption is that the observer reads out the mean of the posterior distribution at the Bayesian decoding stage. This presents the optimal strategy for minimizing the expected squared error (*Ma, 2019*), but it is unclear whether the visual system employs this strategy. It is important to note that different encoding schemes and read-out rules may or may not lead to similar predictions. More work is needed to explore the validity of these assumptions and the consequences for modeling attractive and repulsive history dependencies.

Notably, the effects of adaptation on neural tuning are diverse, and may depend on many stimulus parameters such as stimulus size and duration in non-trivial ways (*Patterson et al., 2014*; *Patterson et al., 2013*). Several neural mechanisms, such as stimulus-specific fatigue and suppressive normalization, have been proposed to underly these neural tuning changes (*Solomon and*

*Kohn, 2014*) and have been related to functional explanations of adaptation, such as efficient coding (*Schwartz et al., 2007*), modulations of stimulus salience (*McDermott et al., 2010*; *Wissig et al., 2013*), and prediction (*Chopin and Mamassian, 2012*). Importantly, the current study cannot explicitly rule out such alternative explanations of repulsive adaptation biases. Rather, here we seek to illustrate a principled way in which the visual system can exploit the temporal stability in the environment across multiple processing stages. Future studies will need to investigate how population-level changes in neural responses may support these different functional explanations of adaptation, respectively.

## Conclusion

Our results demonstrate that visual processing is concurrently influenced by repulsive encoding and attractive decoding biases, operating over distinct timescales. While attractive decoding biases are confined to the immediately preceding stimuli, repulsive encoding biases decay at a distinctly slower rate. The temporal pattern of short-term attraction and long-term repulsion can be well captured by an ideal observer model with efficient encoding and Bayesian decoding of visual information in a stable environment. This model illustrates a principled way in which observers can exploit the temporal stability in the environment across multiple processing stages to concurrently achieve efficient and stable visual processing.

# Materials and methods

## General methods

Participants were recruited from the institute's subject pool and received either monetary compensation or study credits. All participants provided written informed consent prior to the start of the experiments. All experiments were approved by the Radboud University Institutional Review Board (CMO region Arnhem-Nijmegen, The Netherlands; Protocol CMO2014/288). Participants reported normal or corrected-to-normal visual acuity. Stimuli were generated with the Psychophysics Toolbox for MATLAB and were displayed on a 24'' flat panel display (Benq XL2420T, resolution 1920 × 1080, refresh rate: 60 Hz). Participants viewed the stimuli from a distance of 53 cm in a dimly lit room, resting their head on a table-mounted chinrest. All data analyses were performed with MATLAB.

## Experiment 1
### Participants

Twenty-three naïve participants (15 female, age 19–29 years) took part in Experiment 1. The sample size was chosen to achieve approximately 80% power for detecting experimental effects that had least a medium effect size (Cohen's d $\geq$ 0.5) with a one-sided test at an alpha level of 0.05.

### Stimuli and design

The sequence of events within each trial is illustrated in *Figure 1*. Throughout the experiment, a central white fixation dot of 0.3° visual angle diameter was presented on a mid-grey background. Participants were instructed to maintain fixation at all times. First, a randomly oriented Gabor stimulus (windowed sine wave grating) was presented left or right of the fixation dot at 6.5° eccentricity. The side of stimulus presentation was constant within a block, but alternated across separate, interleaved blocks. The Gabor stimulus could either have a low or high spatial frequency (0.33 or 0.5 cycles/°) and had a fixed phase. We introduced the spatial frequency manipulation to investigate a different research question, unrelated to the current investigation of the timescales of positive and negative serial dependence biases, and thus ignored this manipulation in our subsequent analyses. Gabor stimuli were presented at 25% Michelson contrast, and windowed by a Gaussian envelope (1.5° s.d.). After 500 ms, the Gabor stimulus was replaced by a noise patch, presented for 1000 ms, to minimize effects of visual afterimages. The noise patch consisted of white noise, smoothed with a 0.5° s.d. Gaussian kernel and windowed by a Gaussian envelope (1.5° s.d.). Following the offset of the noise patch and a delay of 250 ms, a randomly oriented response bar (0.3° wide white bar windowed by a 0.8° Gaussian envelope) appeared at the same location. The participants' task was to adjust the response bar such that it matched the orientation of the Gabor patch seen just before. Once adjusted to the desired position, the response was submitted by pressing the space bar. The

response was followed by a 2 s inter-trial-interval. Each participant completed a series of 820 trials, divided into 10 blocks, run in a single experimental session.

## Data analysis
### Outlier correction
Prior to analyzing serial dependence biases, we discarded those trials in which the response error (shortest angular distance between stimulus and response orientation) was further than three circular standard deviations away from the participant's mean response error. This was done in order to exclude trials on which the participant gave random responses due to blinks or attentional lapses during stimulus presentation as well as due to inadvertent responses. Subsequently, we subtracted the mean response error from all responses to remove general clockwise or counterclockwise response biases that were independent of the stimulus history. On average, 13.43 ± 9.31 (mean ±s. d.) of 820 trials were excluded per participant and the mean absolute response error was 9.15°±2.47 (mean ±s.d.). Participants took on average 1.63 s to respond.

### Quantifying serial dependence
We quantified the systematic biases of adjustment responses by the stimuli presented on the 40 preceding trials. For the immediately preceding (1-back) trial, response errors were expressed as a function of the difference between previous (1-back) and current stimulus orientation. For positive values of this difference, the previous stimulus was oriented more clockwise than the current stimulus. Similarly, positive response errors denote trials in which the response bar was adjusted more clockwise than the stimulus. Consequently, data points in this error plot (*Figure 2A*) that have x- and y-values of the same sign represent trials in which the response error was in the direction of the previous stimulus orientation. In order to quantify the systematic bias, we pooled the response errors of all participants and fitted the first derivative of a Gaussian curve (DoG, see *Fischer and Whitney, 2014*) to the group data. The DoG is given by $y = xawce^{-(wx)^2}$, where x is the relative orientation of the previous trial, a is the amplitude of the curve peaks, w is the width of the curve and c is the constant $\sqrt{2}/e^{-0.5}$. The constant c is chosen such that parameter a numerically matches the height of the curve peak. The amplitude parameter a was taken as the strength of the serial dependence bias, as it indicates how much the response to the current stimulus orientation could be biased towards or away from a previous stimulus with the maximally effective orientation difference between stimuli. For all model fits, the width parameter w of the DoG curve was treated as a free parameter, constrained to a range of plausible values (w = 0.02–0.07, corresponding to curve peaks between 10° and 35° orientation difference). To assess the systematic influence of n-back stimuli on the current adjustment response, we repeated the above procedure, but conditioned response errors on the difference between the n-back and current stimulus orientation. We only considered trials, which had an n-back stimulus in the same experiment block. Consequently, when assessing 1-back serial dependence, we discarded the first trial of each block. Conversely, for the 40-back analysis, we discarded the first 40 stimuli of each block, that is nearly half of the data. Therefore, biases by temporally more remote stimuli are estimated for a lower number of trials, reflected in the increasing error bars for bias estimates (*Figure 2C*).

### Statistical analysis
We used permutation tests to statistically assess serial dependence biases on the group level, separately for each n-back stimulus. A single permutation was computed by first randomly inverting the signs of each participant's n-back conditioned response errors (i.e. changing the direction of the response errors). This is equivalent to randomly shuffling the labels between the empirically observed data and an artificial null distribution of no serial dependence (a flat surrogate response error distribution) and subtracting the two conditions from each other per participant. Subsequently, we fitted a new DoG model to the pooled group data and collected the resulting amplitude parameter a in a permutation distribution. We repeated this permutation procedure 10,000 times. As p-values we report the percentage of permutations that led to (1) equal or higher values for a than the one estimated on the empirical data, when the empirical estimate of a was positive, or (2) equal or lower values for a than the one estimated on the empirical data, when the empirical estimate of a was negative. As we conducted a two-sided test, we multiplied this p-value by two and set the

significance level to α = 0.05. Subsequently, we Bonferroni-corrected for multiple comparisons, resulting in a significance level of α = 0.00125.

## Quantifying variability of response errors

In order to test whether the variability of response errors was dependent on the relative orientation difference between current and previous stimuli, we computed the standard deviation of response errors in a 30° sliding window over relative orientation differences between current and previous stimulus, separately for each participant. Subsequently, we normalized the resulting conditioned variability estimates by subtracting each participant's mean response variability estimate and averaged the normalized variability estimates across participants. To quantify systematic changes in response variability, we fitted the second derivative of a Gaussian curve (2-DoG) to the group data (*Figure 7— figre supplement 4*). The 2-DoG is given by $y = -a\,(2w^2x^2 - 1)e^{-(wx)^2} + b$, where $x$ is the relative orientation of the previous trial, $a$ is the amplitude of the negative peak at $x = 0$, $w$ controls the width of the curve and $b$ controls the vertical offset. The amplitude parameter $a$ was taken as the strength of the response variability modulation, as it indicates how much the response variability is reduced when current and previous stimulus orientations are identical. For all model fits, the width parameter $w$ of the DoG curve was treated as a free parameter, constrained to a range of plausible values ($w = 0.01 - 0.1$). To assess the systematic influence of n-back stimuli on current response variability, we repeated the above procedure, but conditioned response error variability on the difference between the n-back and current stimulus orientation (*Figure 7—figre supplement 4*, right column).

# Experiment 2

The data of Experiment 2 were collected as part of a previously published study, where we reported on the bias exerted by the previous stimulus (Experiment 1 in *Fritsche et al., 2017*). Here, we reanalyzed this dataset to investigate biases exerted by stimuli encountered further in the past, when presented at the same or different spatial location as the current stimulus.

## Participants

Twenty-five naïve participants took part in Experiment 2. One participant was excluded from data analysis due to difficulties in perceiving peripherally presented stimuli during the experiment. Thus, data of twenty-four participants (19 female, age 18–27 years) were analyzed. This target sample was chosen to achieve approximately 80% power for detecting experimental effects that had at least a medium effect size (Cohen's d $\geq$ 0.5) with a one-sided test at an alpha level of 0.05.

## Stimuli and design

Experiment 2 was similar to Experiment 1, but the location of stimulus presentation could change between upper and lower half of the visual field on a trial-by-trial basis. Throughout the experiment, a central white fixation dot of 0.25° visual angle diameter was presented on a mid-grey background. Participants were instructed to maintain fixation at all times. First, a cue (white disc windowed by a 0.7° Gaussian envelope) appeared at 10° horizontal and 5° vertical eccentricity from fixation for 350 ms. Presentation in the left and right visual field was alternated in separate, interleaved blocks, while presentation in the upper and lower visual field was pseudo-randomized across trials. After further 350 ms of fixation, a randomly oriented Gabor stimulus (windowed sine wave grating) was presented at the same location as the cue. Gabor stimuli were windowed by a Gaussian envelope (1.5° s.d.), had a spatial frequency of 0.33 cycles/° with fixed phase, and were presented at 25% Michelson contrast. After 500 ms, the Gabor stimulus was replaced by a noise patch, presented for 1000 ms, to minimize effects of visual afterimages. The noise patches consisted of white noise, smoothed with a 0.3° s.d. Gaussian kernel, windowed by a Gaussian envelope (1.5° s.d.) and presented at 50% contrast. A response bar (0.6° wide white bar windowed by a 1.3° Gaussian envelope) appeared 250 ms after the offset of the noise patch at the same location. The participants' task was to adjust the response bar such that it matched the orientation of the Gabor stimulus seen just before. Once adjusted to the desired position, the response was submitted by pressing the space bar. The response was followed by a 2 s inter-trial-interval. Each participant completed a series of 808 trials, divided into eight blocks. After each block, participants received feedback about their mean (absolute) response error. The temporal sequence of stimulus presentations in the upper and lower visual field was

counterbalanced with respect to the stimulus location of trial *n* and *n-1*. Consequently, on half of the trials stimuli were presented at the same spatial location as the previous stimulus (same-location trials), while on the other half of the trials the spatial location of current and previous stimulus changed (different-location trials).

## Data analysis
### Outlier correction

We performed the same outlier correction and removal of the general (history-independent) response bias as in Experiment 1. On average, 9.08 ± 6.96 trials (mean ±s.d.) of 808 trials were excluded per participant and the overall mean response error was 7.21 ± 1.45˚ (mean ±s.d.). Participants took on average 2.10 s to respond.

### Quantifying serial dependence

To estimate the systematic biases exerted by past stimuli, we performed the same DoG model fitting procedure as described in Experiment 1. In order to gain statistical sensitivity, we focused on the influence of the past 10 trials, for which Experiment one revealed the strongest repulsive biases. This analysis was performed on all trials, regardless of spatial location. To investigate the spatial specificity of the long-term repulsive bias, we subsequently pooled response errors conditioned on 4- to 9-back trials, which showed the overall strongest repulsive biases in this experiment. We subsequently split the data according to whether the current and 4- to 9-back stimulus were presented at the same or a different spatial location and repeated the model fitting procedure separately for the two subsets of data. This analysis captured how much the current adjustment response was biased on average by a stimulus occurring 4 to 9 trials ago, depending on the spatial overlap between past and current stimulus.

### Statistical analysis

We statistically assessed the biases exerted by the 10 past stimuli, regardless of their spatial location, using the same two-sided permutation test described in Experiment 1. The significance level was first set to $\alpha = 0.05$, and subsequently Bonferroni-corrected for multiple comparisons, resulting in a significance level of $\alpha = 0.005$. To test the spatial specificity of the repulsive bias, we conducted a second permutation test. For each permutation, we randomly shuffled the condition labels of the same-location and different-location condition of the pooled 4- to 9-back conditioned response errors of each participant. We then fitted DoG models to the permuted conditions, pooled across participants, and recorded the difference between the amplitude parameters *a*. We repeated this procedure 10,000 times. As p-values, we report the percentage of permutations that led to an equal or more extreme amplitude difference than the one we observed in the experiment, multiplied by a factor of 2 to reflect the two-sided test. The significance level was set to $\alpha = 0.05$. The exchangeability requirement for permutation tests is met, because under the null hypothesis of no difference in serial dependence between same-location and different-location conditions, the condition labels are exchangeable.

### Quantifying variability of response errors

The variability of response errors was quantified with the same procedure as described in Experiment 1.

## Experiment 3

The data of Experiment 3 were collected as part of a previously published study, where we reported on the bias exerted by the previous stimulus (Experiment 4 in *Fritsche et al., 2017*). Here, we reanalyzed this dataset to investigate biases exerted by stimuli encountered further in the past, when the working memory delay on the current trial was either short or long. Experiment 4 was conducted in two separate sessions. Experimental sessions were conducted on different days, but scheduled no longer than three days apart. The experimental procedures were similar across the two sessions.

## Participants

Twenty-five naïve participants took part in Experiment 3. One participant aborted the experiment during the first session due to nausea symptoms and was consequently excluded from the remainder of the experiment and data analysis. Thus, data of 24 participants (15 female, age 18–30 years) were analyzed. This target sample size was chosen to achieve approximately 80% power for detecting experimental effects that had at least a medium effect size (Cohen's d $\geq$ 0.5) with a one-sided test at an alpha level of 0.05.

## Stimuli and design

The experiment was similar to Experiment 2 and unless otherwise stated, the stimulus parameters were identical. At the beginning of each trial, the fixation dot briefly dimmed for 250 ms to indicate the upcoming appearance of a randomly oriented Gabor stimulus. After further 250 ms, the Gabor was presented at 6.5° horizontal eccentricity from fixation. Presentation in the left and right visual field was alternated in separate interleaved blocks. After 250 ms, the Gabor was replaced by a noise patch, also presented for 250 ms. The offset of the noise patch was followed by either a 50 ms (short response delay) or a 3500 ms (long response delay) period of fixation. Subsequently, a randomly oriented response bar appeared at the same location and participants adjusted the bar's orientation to reproduce the orientation of the Gabor seen just before. Participants were instructed to submit their response by pressing the space bar within a fixed response period of 3000 ms. After submitting the response, the bar disappeared from the screen. When participants failed to respond within the response period, the fixation dot's color briefly changed to red (250 ms). The response period was followed by an inter-trial interval (ITI) with one of two lengths, contingent on the response delay: in short response delay trials the ITI was 4200 ms, whereas in long response delay trials it was 750 ms. This timing, together with the fixed response interval, ensured that the Gabor on the following trial was always presented exactly 8 s after the offset of the Gabor on the current trial, regardless of the length of the response delay. Each participant completed 1212 trials, divided into 12 blocks that were distributed over two sessions. The duration of one session was ~110 min. After each block, participants received feedback about their mean (absolute) response error, averaged over response delays. The temporal sequence of short and long response delays was counterbalanced with respect to the response delay of trial *n* and *n-1*, such that each combination of response delays on the current and previous trial occurred equally often.

## Data analysis

### Outlier correction

We excluded those trials from further analysis in which the response error (shortest angular distance between stimulus and response orientation) was further than three circular standard deviations away from the participant's mean response error. This was done separately for short and long response delay trials in order to prevent a bias towards removing long delay trials, for which response error distributions had a higher variance. Subsequently, we removed general clockwise or counterclockwise response biases that were independent of biases due to stimulus history by removing the mean response error from each participant's responses. For short response delay trials, an average of 9.25 ± 5.78 trials (mean ±s.d.) of 606 trials was excluded per participant. For long response delay trials, 9.58 ± 6.83 trials were excluded. Participants failed to give a response within the response period in 18.38 ± 22.27 trials. Mean (absolute) response errors were significantly larger for long response delay trials than for short response delay trials (long: 9.14°±2.32; short: 7.67°±2.04; t(23) = 8.89, p=6.75 $\times$ 10$^{-9}$; paired t-test), indicating that the working memory representation of the Gabor's orientation deteriorated during the response delay.

### Quantification and statistical analysis of serial dependence

We statistically assessed the biases exerted by the 10 past stimuli, regardless of working memory delay, using the same two-sided permutation test as described in Experiment 1 and 2. The significance level was first set to $\alpha$ = 0.05, and subsequently Bonferroni-corrected for multiple comparisons, resulting in a significance level of $\alpha$ = 0.005. To assess the modulation of the long-term repulsive bias by current working memory delay, we employed a similar procedure as in Experiment 2. First, we defined a time window of interest, ranging from 2- to 6-back trials, for which the

repulsive biases were generally most pronounced. We then pooled the response errors conditioned on stimuli from this time window, and split these data according to the response delay on the current trial. Similar to assessing the difference of serial dependence between same-location and different-location trials in Experiment 2, we assessed the difference of serial dependence between short and long response delay trials with a permutation test (see above). The significance level was set to $\alpha = 0.05$.

A priori, we hypothesized that the long-term repulsion bias would not be modulated by the working memory delay on the current trial. In order to quantify evidence for this null hypothesis, we employed a Bayes Factor analysis. In contrast to the permutation tests described above, the Bayes Factor analysis required us to estimate serial dependence biases on the participant level. In some cases, appropriate DoG model fits to the participant's data could not be obtained, as their data did not strongly follow the DoG model's shape. In order to obtain estimates of serial dependence on the participant level independent of model fit, we used a model free quantification. To this end, we averaged response errors of trials in which the n-back stimulus was oriented more clockwise or counter-clockwise than the current stimulus, respectively. Subsequently, we subtracted the average error on 'n-back counterclockwise' trials from the error on 'n-back clockwise' trials. The resulting estimate was divided by 2, to reflect the participant's average response error in the direction of the n-back stimulus. Positive values indicated an attraction bias. The single participant estimates of 1-back and 2- to 6-back serial dependence were subsequently used for conducting Bayesian t tests with default Cauchy priors (scale 0.707).

Furthermore, to directly compare working memory delay modulations between short-term attractive and long-term repulsive biases (i.e. their interaction), we entered single participant estimates into a repeated measures ANOVA with factors n-back (1-back vs. 2- to 6-back) and working memory delay (short vs. long). As we were interested in the modulation of bias magnitudes, independent of their direction, we entered the absolute values of bias estimates. The significance level was set to $\alpha = 0.05$.

## Quantifying variability of response errors
The variability of response errors was quantified with the same procedure as described in Experiment 1.

## Experiment 4
The data of Experiment 4 were collected as part of a previous study, where we reported on the bias exerted by the history-inducing stimulus on the current (Experiment 2 in *Fritsche et al., 2017*). Here, we reanalyzed this dataset to investigate biases exerted by inducer stimuli encountered further in the past.

### Participants
The same 25 naïve participants, who took part in Experiment 2, participated in Experiment 4. One participant was excluded from data analysis due to problems of seeing peripherally presented stimuli during the experiments. Thus, data of twenty-four participants (19 female, age 18–27 years) were analyzed.

### Stimuli and design
In Experiment 4, participants had to perform two consecutive tasks on each trial (*Figure 5A*). First, participants performed an adjustment task similar to that of Experiment 1, 2 and 3 (adjustment response). Subsequently, two Gabor stimuli were presented (one at the location of the previous adjustment response, and one at a location 20 visual degrees apart) and participants judged which of the two stimuli was tilted more clockwise (two-alternative forced choice, 2AFC). Unless otherwise stated, stimulus parameters where the same as in Experiment 2. First, a cue appeared at 10° horizontal and 5° vertical eccentricity from fixation and informed participants about the location of the stimulus whose orientation they would need to reproduce. Afterwards, two Gabor stimuli were presented – one at the location of the cue, the other in the opposite hemifield – and subsequently masked by two noise patches. Next, a response bar appeared at the location of the cued stimulus and participants had to adjust the bar's orientation to that of the stimulus. The submission of the

response was followed by a 1300 ms period of fixation. The second half of each trial started with two cues simultaneously presented for 350 ms. The cues were either presented at the same locations as the adjustment stimuli (same-location trials), or 10° above or below, respectively (different-location trials). After 350 ms of fixation, two Gabor stimuli were presented at the locations of the cues. Orientations of both Gabor stimuli were constrained to the range of −14.5 and 14.5° around vertical and the orientation difference between the stimuli was varied between −9 to 9° in steps of 3°. After 500 ms the stimuli were masked by noise patches, presented for 1000 ms. After the offset of the noise patches, participants had to indicate which one of the two stimuli was oriented more clockwise (or counterclockwise, instruction was varied in separate blocks) by pressing the left or right arrow key. The response was followed by a 1,500 ms fixation period after which the next trial began. Crucially, the orientations of the first set of stimuli (adjustment stimuli) were chosen as follows. The stimulus at the cued location was tilted either −20 or 20° with respect to the 2AFC stimulus subsequently appearing at the same side of fixation. This orientation difference was chosen, as it has been previously shown to induce maximally attractive serial dependence effects on perceptual choices (*Fischer and Whitney, 2014*). The stimulus at the uncued location had a random orientation in the range of −34.5 and 34.5°, the range in which all stimulus orientations fell. Thus, the stimulus orientation at the uncued location had no systematic relationship with either of the two 2AFC stimuli appearing afterwards. Participants completed 8 blocks of 56 trials each. After each block, participants received feedback about their average (absolute) response error on the adjustment task and accuracy on the 2AFC task. The order of same-location and different-location trials, as well as the tilt direction and vertical position of the adjustment stimulus were counterbalanced and pseudo-randomized across trials. Importantly, while we were previously concerned with the bias exerted by the current inducer stimulus on the current perceptual comparison (2AFC judgment), here we investigated the influence of previous trials' inducer stimuli on the current perceptual comparison. The orientation statistics of the experiment implied that those inducer stimuli of previous trials had relative orientations ranging from −49 to 49° with respect to the current 2AFC stimulus appearing at the same side of fixation.

## Data analysis

The purpose of the adjustment phase of each trial was to introduce a stimulus history, which could subsequently induce a bias in the perceived orientation of the Gabor presented later, at the same or vertically nearby location. Hence, we refer to the adjustment stimulus as the inducer stimulus. While we were previously concerned with the bias exerted by the inducer stimulus presented on the same trial as the subsequent 2AFC stimuli, here we investigated whether inducer stimuli, presented on previous trials, bias the perceptual comparison of the 2AFC stimuli on the current trial. To this end, we binned the data into two bins according to the influence of the n-back inducer stimulus on the current 2AFC response that would be expected under repulsive adaptation. For instance, for a trial in which the 2AFC stimulus in the right visual field was preceded by a more counterclockwise n-back inducer, repulsive adaptation would bias the perceived orientation of the right stimulus in the clockwise direction. Consequently, when judging which 2AFC stimulus was oriented more clockwise, choosing the stimulus in the right visual field would be facilitated. Similarly, for a trial in which the 2AFC stimulus in the left visual field was preceded by a more clockwise n-back inducer, repulsive adaptation would bias the perceived orientation of the left stimulus in the counterclockwise direction, again making it more likely that the participant would judge the stimulus in the right visual field as more clockwise. These two types of trials were combined in the same 'adapt right' bin, since the n-back inducer would facilitate a 'right' response in both cases. Analogously, we collected the trials for which repulsive adaptation would make participants more likely to report the stimulus in the left visual field as more clockwise in a 'adapt left' bin. The binning procedure was performed separately for trials in which the n-back inducer and current 2AFC stimuli were presented at the same vertical location (same location trials) or at a different vertical location (different location trials). In order to quantify the bias of the n-back inducer on perceived orientation of the 2AFC stimuli, we fitted psychometric functions to each participant's data, separately for 'adapt right' and 'adapt left' bins. The psychometric function is given by $\Psi(x; \alpha, \beta, \lambda) = \lambda + (1 - 2\lambda) F(x; \alpha, \beta)$, where $\Psi$ describes the proportion of 'right' responses and $F$ is a cumulative Gaussian distribution. Parameters $\alpha$ and $\beta$ determine the location and slope of the psychometric function and $x$ denotes the orientation difference

between right and left 2AFC stimuli on the current trial. Parameter λ account for stimulus independent lapses and was fixed to 0.01. Importantly, α corresponds to the point of subjective equality (PSE): the orientation difference between right and left stimulus for which participants perceived the orientations as equal. We estimated PSEs for 'adapt right' and 'adapt left' conditions and recorded the difference as $\Delta PSE = PSE_{bias\ left} - PSE_{bias\ right}$, separately for same-location and different-location trials. We quantified the overall bias on perception as 0.5 * ΔPSE. This value indicates how much a 2AFC stimulus is biased by a single inducer stimulus. While positive values signal an attraction effect exerted by the n-back inducer, negative values imply repulsion. This analysis was performed separately for the inducer stimuli of the previous 10 trials, reaching back ~100 seconds into the past. To statistically assess whether previous inducer stimuli biased the current perceptual comparison, we averaged the bias estimates of the past 10 trials per participant, separately for same-location and different-location trials and compared these average biases against zero using two-sided one-sample t-tests (α = 0.05). Furthermore, we compared the average 1- to 10-back bias between same and different location trials using a two-sided paired t-test (α = 0.05). In order, to characterize the decay of the n-back biases over past trials, we fit exponential functions of the form $N(n) = N_0 e^{-\lambda n}$ to the same- and different location bias estimates, where $n$ denotes the index of the n-back trial. We converted the decay constant λ into the half-life $t_{1/2}$ estimate with the formula $t_{1/2} = \frac{\ln(2)}{\lambda}$.

## Ideal observer model

We modeled orientation estimates as resulting from a probabilistic encoding-decoding process. In this process, a physical grating stimulus with orientation $\theta$ elicits a noisy sensory measurement $m$ (encoding), which is subsequently used to generate an estimate $\hat{\theta}(m)$, representing the observer's estimate of the stimulus orientation (decoding). To model history dependencies, we made two assumptions. First, we assumed that the encoding of the physical stimulus is efficient (***Wei and Stocker, 2015***), such that the sensory system is optimally adapted to represent the current stimulus orientation. In particular, in a stable environment in which current visual input is similar to previous input, the prior probability over current stimulus orientations is dependent on previously experienced stimuli. This predictability between previous and current stimuli allows the sensory system to allocate coding resources such that the mutual information between the likely physical stimulus and its sensory representation is maximized. Second, we assumed that at the decoding stage, similar to encoding, the observer forms a prediction about the current stimulus based on knowledge about previous stimuli and combines this prediction with the current sensory measurement in a probabilistically optimal, that is Bayesian manner (***van Bergen and Jehee, 2019***). If predictions are well matched to the temporal regularities in the environment, combining such predictions with current sensory measurements generally produces more accurate stimulus estimates than those based on current sensory measurements alone. Using data of the three orientation estimation tasks (Experiments 1 to 3), we quantitatively compared such an ideal observer model with efficient encoding and Bayesian decoding, to observers that only exhibited efficient encoding or Bayesian decoding.

## Efficient encoding

The current efficient coding implementation largely follows the framework put forward by Wei and Stocker (***Wei and Stocker, 2015***; ***Wei and Stocker, 2017***). Briefly, we assumed that sensory encoding maximizes the mutual information I[θ, m] between the sensory measurement $m$ and the physical stimulus orientation $\theta$ with regard to the intrinsic uncertainty in the sensory representation. To this end, the observer leverages its knowledge about the prior probability distribution over current stimulus orientations $p(\theta)$. By using a tight bound on mutual information, one can link Fisher information $J(\theta)$ of the sensory representation to the prior stimulus probability $p(\theta)$, such that $p(\theta) \propto \sqrt{J(\theta)}$ (for details see ***Wei and Stocker, 2015***). Fisher information reflects the amount of coding resources that is dedicated to the representation of a certain stimulus orientation $\theta$. According to the above formulation, coding resources are allocated such that the most likely stimulus orientation is represented with highest accuracy. In order to specify the encoding model in terms of a likelihood function, we further assumed that sensory noise, and thus the likelihood function, is homogenous and symmetric around the physical stimulus orientation in a sensory space with uniform Fisher information defined by the mapping $F(\theta)$, where $F(\theta)$ is the cumulative prior probability distribution over current stimulus orientations. After defining the likelihood function in sensory space, one can then apply the inverse

mapping $F^{-1}\left(\tilde{\theta}\right)$ to obtain the likelihood in stimulus space: $p(m|\theta)$. This likelihood is typically asymmetric, with a long tail away from the peak of the prior probability distribution (*Figure 6B*; for further intuition about the asymmetry in the likelihood function see *Stocker and Simoncelli, 2006*).

Crucially, in contrast to *Wei and Stocker, 2015*, who assumed a fixed prior probability distribution $p(\theta)$, here we assumed that the sensory system leverages previous sensory measurements $m^{(t_0:t-1)}$, in order to predict the current stimulus $\theta^{(t)}$. In particular, predictions are based on an internal model that approximates the true changes in orientation statistics in natural environments (*van Bergen and Jehee, 2019*):

$$p_{encoding}\left(\theta^{(t)}\,|\,\theta^{(t-1)}\right) = p_{same}\, C\left(\theta^{(t)}, \theta^{(t-1)}, \sigma_{td}\right) + (1 - p_{same})\, U(0, 180)$$

with

$$C\left(\theta^{(t)}, \theta^{(t-1)},\, \sigma_{td}\right) = \frac{1}{Z}\exp\left(-\frac{1}{2\sigma_{td}^2}\, angle\left(\theta^{(t)}, \theta^{(t-1)}\right)^2\right)$$

where Z is a normalization constant that was computed numerically. This transition distribution describes the probability that a stimulus at time $t$ has the orientation $\theta^{(t)}$, given that the previous stimulus had the orientation $\theta^{(t-1)}$. The distribution is formed by a mixture of a central Gaussian peak with standard deviation $\sigma_{td}$, and a uniform distribution. The uniform distribution accounts for cases, in which orientations change suddenly and unpredictably, occurring with probability $(1 - p_{same})$. $t_0$ denotes the starting time point of the inference process.

In order to compute a prediction about the current stimulus orientation, $p\left(\theta^{(t)}\,|\,m^{(t-1)}\right)$, the sensory system then combines this model of natural orientation changes with its knowledge about the previous stimulus orientation, based on the previous sensory measurement:

$$p\left(\theta^{(t)}\,|\,m^{(t-1)}\right) = \int p_{encoding}\left(\theta^{(t)}\,|\,\theta^{(t-1)}\right) p\left(\theta^{(t-1)}\,|\,m^{(t-1)}\right) d\theta^{(t-1)}$$

where

$$p\left(\theta^{(t-1)}\,|\,m^{(t-1)}\right) = \frac{1}{Z}\, p\left(m^{(t-1)}\,|\,\theta^{(t-1)}\right)$$

Here, $p\left(m^{(t-1)}\,|\,\theta^{(t-1)}\right)$ is the likelihood function of the previous stimulus, and Z is a numerically computed normalization constant that ensures that $\int p\left(\theta^{(t-1)}\,|\,m^{(t-1)}\right) = 1$. Since statistical regularities in natural environments do not only exist between immediately consecutive orientation samples, but over extended timescales, we further assumed that the sensory system does not only base its prediction about the current stimulus $\theta^{(t)}$ on the previous measurement, but forms a prediction based on an exponentially weighted mixture of previous predictions:

$$p_{mix}\left(\theta^{(t)}\,|\,m^{(t_0:t-1)}\right) = \sum_{i=0}^{t-1} w_i\, p\left(\theta^{(t-i)}\,|\,m^{(t-i-1)}\right)$$

with weights

$$w_i = \frac{1}{Z}\exp\left(-(t-i)/\tau_{encoding}\right)$$

Where $\tau_{encoding}$ is the exponential integration time constant that determines the time window over which the sensory system integrates previous predictions to predict the current stimulus orientation, and Z was a normalization constant, chosen such that $\sum_{i=0}^{t} w_i = 1$. Here, the exponential decay function is defined over elapsed trials, where $t$ denotes the index of the current trial and $i$ denotes the index of a previous trial. Thus, $\tau_{encoding}$ denotes the number of elapsed trials after which the contribution of a stimulus to the encoding prior has decreased to $1/e \approx 0.37$ of its initial value. From this, we derived a half-life estimate with the formula $t_{1/2} = \tau_{encoding} * \ln(2)$ and converted this estimate to

seconds by multiplying with the average trial duration of the respective experiment (average trial durations: Exp. 1: 5.38 s; Exp. 2: 6.55 s; Exp. 3: 8.25 s).

On a given trial, the sensory system then used the prediction $p_{mix}\left(\theta^{(t)}|\,m^{(t_0:t-1)}\right)$ to allocate sensory resources for the efficient encoding of stimulus $\theta^{(t)}$, as described above, resulting in a new sensory measurement $p\left(m^{(t)}|\,\theta^{(t)}\right)$.

## Bayesian decoding

After obtaining a noisy sensory measurement of the current stimulus, $p\left(m^{(t)}|\,\theta^{(t)}\right)$, the observer seeks to infer the stimulus orientation, which most likely caused the measurement. Importantly, when doing so, the observer does not only rely on the current sensory measurement, but makes use of predictions based on knowledge about previous stimulus orientations (*van Bergen and Jehee, 2019*). While, in principle, encoding and decoding stages could utilize the same predictions about the upcoming stimulus (see '*Encoding-decoding model (single prior)*' below), here we allowed encoding and decoding to be based on separate predictions, involving separate transition models and integration time constants. For instance, transition models in low-level sensory circuits, encoding visual information, could be acquired over the organism's lifetime, whereas higher-level transition models used for decoding could be context dependent (*Fischer et al., 2020*) and learned quickly, within minutes to hours (*Braun et al., 2018*).

Following *van Bergen and Jehee, 2019*, predictions at the decoding stage were computed by convolving the distribution of knowledge about the previous stimulus $p\left(\theta^{(t)}|\,m^{(t_0:t-1)}\right)$ with the distribution of changes that can occur between two consecutive measurements, as given by the transition model $p_{decoding}\left(\theta^{(t)}|\,\theta^{(t-1)}\right)$:

$$p\left(\theta^{(t)}|\,m^{(t_0:t-1)}\right) = \int p_{decoding}\left(\theta^{(t)}|\,\theta^{(t-1)}\right)p\left(\theta^{(t-1)}|\,m^{(t_o:t-1)}\right)d\theta^{(t-1)}$$

Importantly, even though the Bayesian inference process is recursive and consequently the prediction $p\left(\theta^{(t)}|\,m^{(t_0:t-1)}\right)$ theoretically contains information about all sensory measurements since the starting point of the inference process, it has been previously found that such a model does not adequately capture positive serial dependencies beyond the most recent trial (*Kalm and Norris, 2018*). Therefore, when predicting the upcoming stimulus orientation, the current observer does not only rely on the most recent prediction, but utilizes an exponentially weighted mixture of previous predictions (*Kalm and Norris, 2018*):

$$p_{mix}\left(\theta^{(t)}|\,m^{(t_0:t-1)}\right) = \sum_{i=0}^{t-1}w_i\,p\left(\theta^{(t-i)}|\,m^{(t_0:t-i-1)}\right)$$

with weights

$$w_i = \frac{1}{Z}\exp\left(-(t-i)/\tau_{decoding}\right)$$

Similar to encoding, the exponential decay function is defined over elapsed trials, where *t* denotes the index of the current trial and *i* denotes the index of a previous trial. Subsequently, the ideal observer combines this prediction with the current sensory measurement according to Bayes, resulting in a posterior distribution:

$$p\left(\theta^{(t)}|\,m^{(t_0:t)}\right) \propto p\left(m^{(t)}|\theta^{(t)}\right)p_{mix}\left(\theta^{(t)}|\,m^{(t_0:t-1)}\right)$$

We assume that the observer reports the mean of this posterior distribution as their best estimate of the viewed stimulus orientation. Reporting the mean presents the optimal strategy for minimizing the expected squared error (*Ma, 2019*).

It should be noted that in this formulation of the ideal observer, the prior of the current trial consists of a mixture of previous trials' posteriors, from which previous perceptual decisions were estimated, convolved with a transition distribution. Therefore, the likelihood of the current trial is pulled towards previous posteriors, more closely reflecting previous decisions, rather than previous likelihoods, reflecting previous stimuli.

## Models

We compared four different models in their ability to capture the temporal pattern of short-term attraction and long-term repulsion biases in our perceptual estimation experiments. While all of these models consisted of an encoding and decoding stage, they differed in how the sensory history influenced encoding and decoding, respectively:

### Bayesian decoding model with history-independent encoding

The first model implemented history-dependent Bayesian decoding, without efficient encoding, and closely resembles a recent ideal observer model, which successfully modeled 1-back attractive serial dependencies (**van Bergen and Jehee, 2019**). In contrast to the full model presented above, in this model the observer encoded physical stimulus orientations as a symmetric, Gaussian likelihood function centered on the true stimulus orientation:

$$p\left(m^{(t)}|\theta^{(t)}\right) = \frac{1}{Z} \exp\left(-\frac{1}{2\sigma_s^2} angle\left(m^{(t)}, \theta^{(t)}\right)^2\right)$$

where $\sigma_s^2$ is the variance of the sensory noise. Consequently, the encoding of the current stimulus was independent of previously encoded stimuli. The model had four free parameters: $\sigma_s$, $\sigma_{td}$, $p_{same}$ and $\tau_{decoding}$.

### Efficient encoding model with history-independent decoding

The second model implemented efficient encoding of sensory information (**Wei and Stocker, 2015**; **Wei and Stocker, 2017**), without history-dependent Bayesian decoding. Thus, while the likelihood function was influenced by previously experienced stimuli through efficient coding (see section *Efficient encoding*), the decoding was based on the sensory measurement alone (analogous to multiplying the likelihood with a uniform, i.e. history-independent prior):

$$p\left(\theta^{(t)}|m^{(t)}\right) = \frac{1}{Z} p\left(m^{(t)}|\theta^{(t)}\right)$$

where *Z* is a numerically computed normalization constant that ensured that $\int p\left(\theta^{(t)}|m^{(t)}\right) = 1$. In this model, the decoding of the current sensory measurement was thus independent of previously experienced stimuli. The model had four free parameters: $\sigma_s$, $\sigma_{td}$, $p_{same}$ and $\tau_{encoding}$.

### Efficient-encoding-Bayesian-decoding model (single prior)

A third model implemented both efficient encoding and history-dependent Bayesian decoding, and utilized the same prior probability distribution over current stimulus orientations, $p_{mix}\left(\theta^{(t)}|m^{(t_0:t-1)}\right)$, both for efficient encoding and Bayesian decoding. Consequently, this observer had a single transition model, $p_{decoding}\left(\theta^{(t)}|\theta^{(t-1)}\right)$, and integration time constant $\tau$ that determined the time window over which the observer integrated previous predictions to predict the current stimulus orientation. The model had four free parameters: $\sigma_s$, $\sigma_{td}$, $p_{same}$ and $\tau$.

### Full efficient-encoding-Bayesian-decoding model

Finally, the full encoding-decoding model implemented efficient encoding and Bayesian decoding, as described above. In this model, efficient encoding and Bayesian decoding could be based on distinct predictions, characterized by different transition models and separate integration time constants $\tau_{encoding}$ and $\tau_{decoding}$, respectively. The model had seven free parameters: $\sigma_s$, $\sigma_{td}$ and $p_{same}$ of the encoding transition model, $\sigma_{td}$ and $p_{same}$ of the decoding transition model, and $\tau_{encoding}$ and $\tau_{decoding}$.

## Model fits

We fitted all four models to the empirical serial dependence biases of Experiment 1 to 3, respectively. In particular, models were fit to the 1- to 20-back group-average serial dependence curves of each experiment. In a first step, we performed a grid search over a wide range of parameter combinations and evaluated the sum of squared residuals (SSR) between predicted and observed serial dependence curves. Subsequently, we used the parameter combination with the lowest SSR as a

starting point for further numerical minimization of SSRs using MATLAB's lsqcurvefit algorithm. To ensure plausible parameter estimates we constrained parameters to $0 < \sigma_s < 60$, $\sigma_{td} > 0$, $0 \leq p_{same} \leq 1$, $\tau_{encoding} > 0$, $\tau_{decoding} > 0$. Parameters of the encoding and decoding transition models of the full encoding-decoding model were constrained to the same ranges.

## Model evaluation

In order to evaluate the models' ability to capture the empirically observed serial dependence biases, we employed a cross-validation procedure. Separately for each experiment, we split the data of each participant into two halves (odd- versus even-numbered blocks) and computed the group-average 1- to 20-back serial dependence curves for each half. We then conducted the model fitting on the first half and testing on the second half. The models' performance was evaluated based on the Pearson correlation between the predicted and observed 1- to 20-back serial dependence curves of the testing dataset. This cross-validation procedure accounts for the increased flexibility of the full encoding-decoding model, due to its greater number of free parameters.

In order to establish a baseline level of chance prediction accuracy of each model, we employed a permutation procedure. Specifically, we randomly permuted the models' predicted responses to the stimuli in the testing dataset, thereby abolishing the trial correspondence between model and participant errors, and computed the correlation between the resulting permuted model predictions and observed serial dependence curves. This procedure was repeated 1000 times and p-values were taken as the proportion of permutation correlations, which exceeded the true correlation between predicted and observed serial dependence curves.

## Additional information

### Competing interests

Floris P de Lange: Senior editor, *eLife*. The other authors declare that no competing interests exist.

### Funding

| Funder | Grant reference number | Author |
| --- | --- | --- |
| H2020 European Research Council | ERC Starting Grant 678286 'Contextvision' | Matthias Fritsche Floris P de Lange |
| Nederlandse Organisatie voor Wetenschappelijk Onderzoek | NWO Veni grant 016. Veni.198.065 | Eelke Spaak |

The funders had no role in study design, data collection and interpretation, or the decision to submit the work for publication.

### Author contributions

Matthias Fritsche, Conceptualization, Resources, Data curation, Software, Formal analysis, Validation, Investigation, Visualization, Methodology, Writing - original draft, Project administration, Writing - review and editing; Eelke Spaak, Formal analysis, Methodology, Writing - review and editing; Floris P de Lange, Conceptualization, Formal analysis, Supervision, Funding acquisition, Writing - review and editing

### Author ORCIDs

Matthias Fritsche (iD) https://orcid.org/0000-0001-5835-9057
Eelke Spaak (iD) http://orcid.org/0000-0002-2018-3364
Floris P de Lange (iD) https://orcid.org/0000-0002-6730-1452

### Ethics

Human subjects: The study followed institutional guidelines of the local ethics committee (CMO region Arnhem-Nijmegen, The Netherlands; Protocol CMO2014/288), including informed consent of all participants.

**Decision letter and Author response**

Decision letter https://doi.org/10.7554/eLife.55389.sa1
Author response https://doi.org/10.7554/eLife.55389.sa2

---

## Additional files

### Supplementary files

• Transparent reporting form

### Data availability

All data and code are openly available on the Donders Institute for Brain, Cognition and Behavior repository at https://doi.org/10.34973/hcea-dt25.

The following dataset was generated:

| Author(s) | Year | Dataset title | Dataset URL | Database and Identifier |
|---|---|---|---|---|
| Fritsche M, Spaak E, de Lange FP | 2020 | A Bayesian and efficient observer model explains concurrent attractive and repulsive history biases in visual perception | https://doi.org/10.34973/hcea-dt25 | Donders Repository, 10.34973/hcea-dt25 |

The following previously published dataset was used:

| Author(s) | Year | Dataset title | Dataset URL | Database and Identifier |
|---|---|---|---|---|
| Fritsche MF, Mostert P, de Lange FP | 2017 | Opposite Effects of Recent History on Perception and Decision | https://doi.org/10.34973/yzm7-bd26 | Donders Repository, 10.34973/yzm7-bd26 |

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
