## [Decision Letter]

Thank you for submitting your article "A Bayesian and efficient observer model explains concurrent attractive and repulsive history biases in visual perception" for consideration by *eLife*. Your article has been reviewed by two peer reviewers, and the evaluation has been overseen by a Reviewing Editor and Joshua Gold as the Senior Editor. The following individual involved in review of your submission has agreed to reveal their identity: Timothy Sheehan (Reviewer #1).

The reviewers have discussed the reviews with one another and the Reviewing Editor has drafted this decision to help you prepare a revised submission.

As the editors have judged that your manuscript is of interest, but as described below that additional analyses are required before it is published, we would like to draw your attention to changes in our policy on revisions we have made in response to COVID-19 (https://elifesciences.org/articles/57162). First, because many researchers have temporarily lost access to the labs, we will give authors as much time as they need to submit revised manuscripts. We are also offering, if you choose, to post the manuscript to bioRxiv (if it is not already there) along with this decision letter and a formal designation that the manuscript is 'in revision at *eLife*'. Please let us know if you would like to pursue this option.

In addition to all of the specific comments from the reviews, we especially focused on the following issues during consultation.

The clarity of model needs to be improved, both in terms of presentation and justifying choices as well as strengthening the link between the Bayesian decoding framework fit and other observations in the literature (e.g. that serial dependence emerges and increases with working memory delay period).

Both reviewers also focused on the importance of accounting for motor biases, which has been shown to be an important contributing factor in response bias for trial history effects (e.g. Akrami et al., 2018). See their comments for more specifics.

During consultation, the reviewers also raised the importance of considering the variability of the discrimination threshold. The current model is based on an efficient coding formulation from Wei and Stocker, which predicts that the discrimination threshold should be inversely proportional to the prior distribution. The authors could re-analyze the data to test this prediction, as least qualitatively.

Reviewer #1:

This manuscript clearly describes a previously underreported perceptual effect including many critical details such as time course and spatial specificity. Reported effects were robust and were replicated in two additional datasets. Fritsche et al. make sense of this phenomenon by building off of recent modeling work utilizing both principals of both Bayesian inference and efficient coding. This model is innovative by combining distinct concepts and introducing an exponential decay of trial influence across time. Critically, it provides a great fit to the pooled data both subjectively (Figure 7—figure supplement 1) and when quantified using cross validation (Figure 7—figure supplement 3). Overall this proposed model adds needed clarity to an area of research with much confusion and apparently contradictory results.

One point of concern is the lack of evidence for efficient encoding being the mechanism leading to repulsion. The authors do not cover an important feature (and the whole theoretical motivation for efficient encoding), that responses are more accurate (or discrimination thresholds smallest) for orientations where Fisher information is highest (see Stoker and Simoncelli, 2006: Figure 4). Since Fisher Information is directly linked to the prior probability, this would be supported by seeing a reduction is absolute error on trials with previous stimuli that are similar in orientation. In line with the exponential decay of efficient encoding, the magnitude of this change in response accuracy (perhaps parameterized with the second derivative of gaussian) should decay with time. If this does not hold true, then the authors should acknowledge other possible explanations for repulsion of previous stimuli including sensory adaptation from, for example, synaptic fatigue (see Solomon and Kohn, 2014).

Reviewer #2:

The authors examined historical effect in perception using data from one new experiment and three previously published datasets. They found both attractive serial dependence (Fischer and Whitney, 2014) and repulsive effect in the same experiments, and crucially the two effects have different time scales. To explain these findings, they modified an efficient coding-bayesian decoding framework (Wei and Stocker, 2015; 2017) and found that a modified model could fit the data well. The new ingredient of the current model is that the predictive prior used in Bayesian inference and efficient coding do not match each other. I think this work contains some interesting results, and could potentially help unify an array of previously disconnected findings. Having said that, I do find that various interpretations of the results to be problematic, and the presentation of the models to be very confusing.

1) Potential confounding factor, i.e., the motor biases, in the reproduction task.

I think it is important to rule this out, in particular for the attractive biases under short time scale. The results in Experiment 4 addresses the perceptual aspect of the long-term repulsive bias to some extent. However, I am puzzled why attractive biases were not observed/reported in this 2-AFC paradigm. If the serial dependence is a perceptual effect, shouldn't we expect attractive biases at short-term scale? I had a difficult time reconciling the results in the two paradigms.

2) The calculation of the historical effect:

Should the attractive/repulsive bias be considered with respect to the orientation of the stimulus or the reported orientation? It would be useful to run the analysis using the reported orientation. Barbosa and Compte (2020, bioRxiv) reports the serial dependence is stronger with using reported stimuli. It would be useful to check whether that's also the case in the authors' data.

If the historical effect need depends on the reported orientation, there seems to be a following-up concern. This one is perhaps naive but could be potentially important if it's true: could attraction biases (toward reported stimulus value) at the short-time scale automatically lead to repulsion at longer time-scale? It would be useful to simulate a ground-truth model with just attraction toward the reported (but no generic repulsion) to see if using the authors' analysis procedure would lead to repulsive effect in longer time scale.

3) The spatial dependence of repulsion and attraction:

The authors claim the attractive biases are not spatially specific (Figure 3), while the repulsive biases are spatially specific. The spatial specificity of the repulsive bias is interpreted as to be consistent with the adaptation effect as measured previously. I found this to be problematic. The two stimulus locations are separated by 13° eccentricity, yet there is still a clear repulsive effect for the "different location" condition. I'd think classical orientation adaptation would lead to almost zero after-effect when the stimulus was to presented to that far away from the adaptor.

4) The role of the noise patch in the experiments is obscure.

If removing the noise patch and using a blank screen instead, would one still observe similar effect? The computational models do not model the noise patch, so I'd think it is fair to say that the model should predict that removing the noise patch would not change any of these biases.

Alternatively, perhaps the noise patch does play a rule. In Experiment 4, noise patch was not used, and interestingly the attractive bias was not observed. So could the noise patch be the main reason why different effects were observed for the two paradigms (this also relates back to my first concern)?

5) The explanation/presentation of the model is highly confusing.

The labeling of efficient coding model and Bayesian decoding model is particularly mis-leading. Each of the models consider in this paper has an efficient encoding component and a Bayesian decoding component. So I found it is strange to label one as encoding model and another as decoding model.

Relatedly, in the sixth paragraph of the subsection “Repulsive and attractive biases can be explained by efficient encoding and optimal decoding of visual information”, the model the authors referring to is a model consisting efficient coding and Bayesian decoding with mismatched prior. It is the mis-match between the encoding and decoding that leads to the explanation power.

Again it is misleading to talk about model with "only Bayesian decoding" or with "only efficient coding". I feel that it is not what actually happens in the models, unless I am profoundly confused with what the authors actually did.

These are some of the instances in the paper which could lead to profound confusions. I'd suggest the authors to systematically re-write the last section of the Results and the related section in the Discussion to make it clear.

6) The dependence of historical biases on time scales within a single experiment has been reported previously. For example, Dekel and Sagi, 2015, reported a sign reversal of the biases when examining different time-scales in adaptation experiments using natural stimuli (their Figure 4). Interestingly, they found the opposite pattern, i.e. repulsion at short-time scale, attraction at longer-time scale. There might be other related studies I missed. I feel that the authors should discuss the relevant findings more thoroughly.

Reference:

Barbosa, João, and Albert Compte. "Build-up of serial dependence in color working memory." bioRxiv (2019): 503185.

---

## [Author Response]

[…] In addition to all of the specific comments from the reviews, we especially focused on the following issues during consultation.The clarity of model needs to be improved, both in terms of presentation and justifying choices as well as strengthening the link between the Bayesian decoding framework fit and other observations in the literature (e.g. that serial dependence emerges and increases with working memory delay period).

We now improved the clarity of the model and strengthened the link between the Bayesian decoding framework and other observations in the literature (subsection “Repulsive and attractive biases can be explained by efficient encoding and Bayesian decoding of visual information”, fifth paragraph). Please see the point-by-point replies to the reviewers’ comments for detailed descriptions of the improvements.

Both reviewers also focused on the importance of accounting for motor biases, which has been shown to be an important contributing factor in response bias for trial history effects (e.g. Akrami et al., 2018). See their comments for more specifics.

We appreciate this concern. However, due to the specific design of the current experiments and empirical evidence provided by previous studies, we regard a systematic contribution of motor biases in the current experiments as highly unlikely. We provide a detailed explanation in our response to point 1 by reviewer 2 and added a discussion of motor biases to the manuscript (subsection “Repulsive and attractive biases can be explained by efficient encoding and Bayesian decoding of visual information”, eighth paragraph).

During consultation, the reviewers also raised the importance of considering the variability of the discrimination threshold. The current model is based on an efficient coding formulation from Wei and Stocker, which predicts that the discrimination threshold should be inversely proportional to the prior distribution. The authors could re-analyze the data to test this prediction, as least qualitatively.

We now re-analyzed the data to test this prediction (subsection “Repulsive and attractive biases can be explained by efficient encoding and optimal 341 decoding of visual information”, seventh paragraph and Figure 7—figure supplement 4). While we indeed confirm such a pattern for the 1-back stimulus, the estimates of sensitivity conditioned on temporally distant trials were relatively noisy and did not provide clear evidence for or against this hypothesis. We now discuss this in the limitations section of the manuscript, and emphasize the need for future studies to test this prediction of the efficient encoding model. Please see point 1 by reviewer 1 for a detailed response to this point.

Reviewer #1:[…] One point of concern is the lack of evidence for efficient encoding being the mechanism leading to repulsion. The authors do not cover an important feature (and the whole theoretical motivation for efficient encoding), that responses are more accurate (or discrimination thresholds smallest) for orientations where Fisher information is highest (see Stoker and Simoncelli, 2006: Figure 4). Since Fisher Information is directly linked to the prior probability, this would be supported by seeing a reduction is absolute error on trials with previous stimuli that are similar in orientation. In line with the exponential decay of efficient encoding, the magnitude of this change in response accuracy (perhaps parameterized with the second derivative of gaussian) should decay with time. If this does not hold true, then the authors should acknowledge other possible explanations for repulsion of previous stimuli including sensory adaptation from, for example, synaptic fatigue (see Solomon and Kohn, 2014).

The current efficient coding scheme in which the observer maximizes the mutual information between physical stimuli and sensory representations would indeed predict that discrimination thresholds should be lowest when previous stimuli were of similar orientation, in line with previous findings of improved discriminability following adaptation (Clifford et al., 2001; Mattar et al., 2018; Regan and Beverley, 1985). In the context of the current estimation tasks, note that we can distinguish two sources of response errors: bias and variability. The original manuscript already describes bias in detail (and reports results in line with the theory; see Figure 7). It is an interesting further prediction of the efficient encoding framework that *sensitivity* should be highest when the Fisher information is highest. This translates to decreased variability of estimation response errors when the current trial has a similar orientation to the previous one. In order to test this prediction, we computed the standard deviation of the response error distribution as a function of the orientation difference between current and n-back trials. We observed that current response variability was markedly decreased when current and 1-back stimuli had similar orientations, and this pattern was well captured by a second-derivative-of-a-gaussian curve (see Figure 7—figure supplement 4). While a similar pattern was observable for the 2-back trial in Experiment 1 and 2, we did not observe a consistent influence of temporally more distant trials across all three experiments. Notably, the estimates of response variability conditioned on temporally distant trials were relatively noisy and could greatly fluctuate from one n-back trial to the next. This might have obscured potentially smaller reductions in response variability conditioned on temporally more distant trials. We thus indeed verify the prediction of the efficient encoding model that responses are more accurate (i.e., less variable, in addition to being less biased) with increasing similarity between successive trials. However, we do not find evidence for a slow exponential decay in variability reduction, as we found in bias. Taken together, we believe that these new results on response error variability neither contradict, nor provide additional support for, the involvement of the efficient coding scheme based on Fisher information. More specific experiments will be necessary to investigate whether long-term repulsion biases are accompanied by reductions in discrimination thresholds. We now report the analysis of response variability in the manuscript (Results section and Materials and methods; Figure 7—figure supplement 4) and discuss this somewhat undecisive additional evidence for the current efficient coding scheme in the limitations section of the Discussion. Furthermore, we now also discuss other forms of adaptation that could underlie the long-term repulsion biases in the Discussion section:

“Notably, the effects of adaptation on neural tuning are diverse, and may depend on many stimulus parameters such as stimulus size and duration in non-trivial ways (Patterson et al., 2014, 2013). […] Future studies will need to investigate how population-level changes in neural responses may support these different functional explanations of adaptation, respectively.”

Reviewer #2:The authors examined historical effect in perception using data from one new experiment and three previously published datasets. They found both attractive serial dependence (Fischer and Whitney, 2014) and repulsive effect in the same experiments, and crucially the two effects have different time scales. To explain these findings, they modified an efficient coding-bayesian decoding framework (Wei and Stocker, 2015; 2017) and found that a modified model could fit the data well. The new ingredient of the current model is that the predictive prior used in Bayesian inference and efficient coding do not match each other. I think this work contains some interesting results, and could potentially help unify an array of previously disconnected findings. Having said that, I do find that various interpretations of the results to be problematic, and the presentation of the models to be very confusing.1) Potential confounding factor, i.e., the motor biases, in the reproduction task.I think it is important to rule this out, in particular for the attractive biases under short time scale. The results in Experiment 4 addresses the perceptual aspect of the long-term repulsive bias to some extent.

We agree with the reviewer that it is important to rule out potential confounding factors, such as motor biases. However, we regard the involvement of systematic motor biases in the reproduction task as highly unlikely. First, in each trial of the estimation tasks (Experiments 1-3), the initial orientation of the response bar was randomly selected from the range of all possible orientations (0,180], and thus the motor actions required for adjusting the response bar to a particular orientation were independent across trials. Second, several previous studies have shown that an explicit reproduction response is not necessary to elicit an attraction bias on the next trial (Czoschke et al., 2019; Fischer and Whitney, 2014; Suárez-Pinilla et al., 2018), that perceptual decisions are attracted towards the presented stimulus orientation rather than a mirrored response orientation (Cicchini et al., 2017), and that attraction biases occur even when performing a different task (and response) on the previous stimulus (Fritsche and de Lange, 2019). Together, these studies strongly suggest that short-term attractive serial dependencies are not due to low-level motor biases. Furthermore, it seems unlikely that low-level motor biases would lead to larger attraction biases with increasing working memory delay (Experiment 3) and that they would induce *spatially specific* long-term repulsion biases (Experiments 2 and 4). As the reviewer already suggests, Experiment 4 provides similar evidence against an involvement of motor biases in the long-term repulsion effect. In this experiment, participants alternate between a reproduction task and 2-AFC task, which require very different motor responses. Furthermore, stimuli in Experiment 4 are counterbalanced such that simple motor biases in the 2-AFC task (e.g. a response to a previous leftward tilted stimulus in the reproduction task induces a bias to press the left response button in the 2-AFC task) would not induce systematic biases in our analysis of the perceptual comparison (2-AFC) data. We now explicitly discuss the evidence against a contribution of motor biases to the short-term attractive and long-term repulsive serial dependencies in the manuscript (subsection “Repulsive and attractive biases can be explained by efficient encoding and Bayesian decoding of visual information”, eighth paragraph).

However, I am puzzled why attractive biases were not observed/reported in this 2-AFC paradigm. If the serial dependence is a perceptual effect, shouldn't we expect attractive biases at short-term scale? I had a difficult time reconciling the results in the two paradigms.

The data of Experiment 4 (2-AFC paradigm) were previously published in Fritsche et al., 2017, in which we investigated the perceptual nature of the short-term attractive serial dependence bias. In this study, we argued that the absence of the attractive bias in perceptual comparison judgments (2-AFC) suggests that the attractive bias is not a perceptual, but a post-perceptual decisional bias (in line with the reviewer’s argument above; but importantly not a motor bias, as argued above). The evidence for a post-perceptual bias was further corroborated by the increase of bias with increasing working memory delay (Experiment 3), pointing towards working memory as the locus of the attraction effect. Importantly, our explanation of attractive serial dependence arising from high-level Bayesian decoding is consistent with a view in which the attractive bias arises during decoding of working memory information. In particular, Bayesian decoding posits that the attraction bias is dependent on the relative widths of sensory likelihood and prior. When sensory likelihoods are narrow, briefly after the initial encoding of the sensory stimulus, the prior has little influence on the resulting posterior, leading only to small attraction biases. As the sensory representation degrades during working memory retention, and the likelihood becomes increasingly broader, the influence of the prior increases. As a result, the posterior will be increasingly pulled towards the prior, leading to an increase in attraction during working memory retention. Importantly, in Experiment 4, observers could compare the 2-AFC stimuli while they were simultaneously presented on the screen, minimizing or even eliminating the influence of working memory noise, and presumably leading to relatively narrow sensory likelihoods at the time of the perceptual decision. As outlined above, we would predict a strong reduction or even absence of attraction biases in this scenario, consistent with the absence of attraction biases in Experiment 4. We now clarified this point in the Discussion section (subsection “Repulsive and attractive biases can be explained by efficient encoding and Bayesian decoding of visual information”, fifth paragraph).

2) The calculation of the historical effect:Should the attractive/repulsive bias be considered with respect to the orientation of the stimulus or the reported orientation? It would be useful to run the analysis using the reported orientation. Barbosa and Compte (2020, bioRxiv) reports the serial dependence is stronger with using reported stimuli. It would be useful to check whether that's also the case in the authors' data.

We thank the reviewer for this suggestion. However, directly conditioning empirical responses on the previous response orientation, rather than the previous stimulus orientation, is problematic, as it can introduce a confound due to history-independent biases, such as the oblique bias (Tomassini et al., 2010). As a consequence, spurious serial-dependence-like bias patterns emerge, even when conditioning on *future* responses, which could not have influenced the current response (see Pascucci et al., 2019, Figure 2F for a demonstration).

Since the response-conditioned serial dependence curve is a superposition of genuine serial dependencies plus a spurious confound, the curve is of higher amplitude than the stimulus-conditioned bias curve, leading to stronger serial dependence estimates. This confound is also present in the preprint by Barbosa and Compte – conditioning on future responses leads to a strong artifactual attraction bias (personal communication with Joao Barbosa). This confound makes it problematic to directly compare response- to stimulus-conditioned serial dependence curves.

Furthermore, as we demonstrate in the current study, any previous response itself is affected by stimuli experienced over minutes into the past. Therefore, these responses will themselves carry information about the history, and will be statistically co-dependent with previous responses. When conditioning response errors on previous responses, this co-dependence between responses may obfuscate the individual contribution of each trial to the current response – a problem which is not present when conditioning on statistically independent stimulus orientations.

Since conditioning on previous responses is problematic, we decided to report stimulus-conditioned response errors, which do not suffer from the above confounds. However, we do believe that current stimulus estimates are biased towards previous stimulus estimates, and not the physical orientations of previous stimuli. This is incorporated in our Bayesian ideal observer model in which sensory representations are integrated with the posteriors of previous trials (see below).

If the historical effect need depends on the reported orientation, there seems to be a following-up concern. This one is perhaps naive but could be potentially important if it's true: could attraction biases (toward reported stimulus value) at the short-time scale automatically lead to repulsion at longer time-scale? It would be useful to simulate a ground-truth model with just attraction toward the reported (but no generic repulsion) to see if using the authors' analysis procedure would lead to repulsive effect in longer time scale.

In our Bayesian decoding model, the prior on the current trial is composed of a mixture of previous trials’ posteriors, from which previous perceptual decisions were derived. Therefore, our model implements an observer in which the current perceptual decision is pulled towards previous perceptual decisions, in line with the model that the reviewer describes above. We find that such a model cannot explain long-term repulsion effects (Figure 7—figure supplement 2, Bayesian decoding only). We now clarified that in the Bayesian decoding model, the likelihood is pulled towards previous posteriors, from which previous perceptual decisions were derived, rather than previous likelihoods, reflecting previous physical stimulus orientations (subsection “Models”).

3) The spatial dependence of repulsion and attraction:The authors claim the attractive biases are not spatially specific (Figure 3), while the repulsive biases are spatially specific. The spatial specificity of the repulsive bias is interpreted as to be consistent with the adaptation effect as measured previously. I found this to be problematic. The two stimulus locations are separated by 13° eccentricity, yet there is still a clear repulsive effect for the "different location" condition. I'd think classical orientation adaptation would lead to almost zero after-effect when the stimulus was to presented to that far away from the adaptor.

There is indeed a small, but statistically significant repulsion bias when test stimuli are presented 10 visual degrees away from the previous stimulus (Experiment 2). Similarly, in Experiment 4, we find a small, but significant repulsion bias when the current inducer stimulus is presented 10 visual degrees away from the subsequent test stimulus. Both findings are consistent with previous studies, which show that weak adaptation effects can occur at spatial locations further away from the adaptor stimulus (e.g. see Knapen et al., 2010, Figure 3A, also using a spatial separation of 10 visual degrees; see also Mathôt and Theeuwes, 2013, Figure 2 “control condition”). We therefore do not regard the smaller adaptation biases in the “different location” conditions in disagreement with previous studies on classical adaptation. We now discuss this in the revised manuscript (subsection “Experiment 2: Long-term repulsive biases are spatially specific”). Crucially, we find a clear difference in repulsion bias between the “same location” and “different location” conditions, both in Experiment 2 and 4. So, even though the repulsion bias is non-zero between different spatial locations, it still clearly is spatially specific. Importantly, we found no difference between “same location” and “different location” for the 1-back attraction effect (Experiment 2), which indicates a spatially unspecific phenomenon.

4) The role of the noise patch in the experiments is obscure.If removing the noise patch and using a blank screen instead, would one still observe similar effect? The computational models do not model the noise patch, so I'd think it is fair to say that the model should predict that removing the noise patch would not change any of these biases.Alternatively, perhaps the noise patch does play a rule. In Experiment 4, noise patch was not used, and interestingly the attractive bias was not observed. So could the noise patch be the main reason why different effects were observed for the two paradigms (this also relates back to my first concern)?

We would like to note that all experiments involved the presentation of noise patches. Due to limited space, we omitted the depiction of noise patches from Figure 5A (Experiment 4). We apologize for the confusion. We now adapted Figure 5A to show that Gabor stimuli were followed by noise patches. As discussed above, we believe the most likely explanation for the differences between Experiment 4 and Experiments 1-3 is that observers had access to both to-be-compared stimuli simultaneously in Experiment 4, thereby eliminating (or strongly reducing) the role of working memory.

5) The explanation/presentation of the model is highly confusing.The labeling of efficient coding model and Bayesian decoding model is particularly mis-leading. Each of the models consider in this paper has an efficient encoding component and a Bayesian decoding component. So I found it is strange to label one as encoding model and another as decoding model.

We apologize for the confusion in presentation. The reviewer is correct in noting that all models considered in the current study consist of an encoding and decoding stage, and we regret not being clearer with our terminology. The differences between the models arise from the different influences of the sensory history on encoding and decoding, respectively.

In the model formerly labelled as the “Bayesian decoding model”, history-dependent prior information is integrated with current sensory information at the decoding stage, while the model encodes physical stimulus orientations as a symmetric, Gaussian likelihood functions centered on the true stimulus orientations. Consequently, the encoding of the current stimulus is independent of the sensory history and thus is not efficient. Therefore, while the model has an encoding stage, it does not have an *efficient* encoding stage. We now labelled this model “Bayesian decoding model with history-independent encoding”.

In model formerly labelled as the “Efficient encoding model”, encoding is dependent on the sensory history, whereas decoding is history-independent. In this model, the stimulus estimate is based on the sensory measurement alone, which is analogous to multiplying the likelihood with a uniform, i.e. history-independent prior. We now labelled this model “Efficient encoding model with history-independent decoding”.

In the models formerly labelled as “Encoding-decoding models”, the sensory history influences both encoding and decoding, and thus they constitute models with both *efficient* encoding and *history-dependent Bayesian* decoding. We now labelled these models “Efficient-encoding-Bayesian-decoding models”.

Additionally, we have now clarified the influence of the sensory history on encoding and decoding stages of the respective models throughout the Results section. In particular, we clarified that in the Efficient encoding model with history-independent decoding there is no influence of the sensory history on the decoding stage (subsection “Repulsive and attractive biases can be explained by efficient encoding and optimal decoding of visual information”, fourth paragraph), whereas in the Bayesian decoding model with history-independent encoding there is no influence of the sensory history on the encoding stage (fifth paragraph of the aforementioned subsection). Furthermore, we now refer to observers in which sensory history influenced only Bayesian decoding or only efficient encoding, rather than models with only Bayesian decoding or only efficient encoding, to avoid confusion about the presence of encoding and decoding stages in the model.

Relatedly, in the sixth paragraph of the subsection “Repulsive and attractive biases can be explained by efficient encoding and optimal decoding of visual information”, the model the authors referring to is a model consisting efficient coding and Bayesian decoding with mismatched prior. It is the mis-match between the encoding and decoding that leads to the explanation power.

The reviewer is correct. We explicitly describe this in the Results section:

“Importantly, we allowed predictions used for optimizing encoding and decoding to be based on potentially distinct transition distributions and integration time constants. This was motivated by the possibility that encoding transition models in low-level sensory circuits could be relatively inflexible and learned over the organism’s lifetime, whereas higher-level transition models used for decoding are likely context- and task-dependent (Fischer et al., 2019) and learned quickly, within minutes to hours (Braun et al., 2018).”

And compare this model to a model with a single prior for efficient encoding and Bayesian decoding:

*“*Furthermore, the observer captured estimation biases far better than an observer in which efficient encoding and Bayesian decoding were based on the same predictions, using the identical transition distribution and integration time constant (see “Efficient-encoding-Bayesian-decoding model (single prior)” in Materials and methods; cross-validated prediction accuracy, Experiment 1: *r* = 0.58 vs. 0.29; Experiment 2: *r* = 0.53 vs. 0.20; Experiment 3: *r* = 0.66 vs. 0.42).”

We also present and discuss the differences between encoding and decoding transition models and time constants in the Results section (subsection “Repulsive and attractive biases can be explained by efficient encoding and optimal decoding of visual information”, sixth paragraph) and the Discussion (subsection “Repulsive and attractive biases can be explained by efficient encoding and Bayesian decoding of visual information”, third paragraph).

Again it is misleading to talk about model with "only Bayesian decoding" or with "only efficient coding". I feel that it is not what actually happens in the models, unless I am profoundly confused with what the authors actually did.

We now improved the description of the models. In particular, we now refer to observers in which sensory history influenced only Bayesian decoding or only efficient encoding (subsection “Repulsive and attractive biases can be explained by efficient encoding and optimal decoding of visual information”), rather than models with only Bayesian decoding or only efficient encoding, to avoid confusion about the presence of encoding and decoding stages in the model. (Both stages are always present, and the investigated models vary with the specific *forms* of encoding and decoding they employ.)

These are some of the instances in the paper which could lead to profound confusions. I'd suggest the authors to systematically re-write the last section of the Results and the related section in Discussion to make it clear.

We now adapted the manuscript to avoid confusions regarding the ideal observer model (see responses above).

6) The dependence of historical biases on time scales within a single experiment has been reported previously. For example, Dekel and Sagi, 2015, reported a sign reversal of the biases when examining different time-scales in adaptation experiments using natural stimuli (their Figure 4). Interestingly, they found the opposite pattern, i.e. repulsion at short-time scale, attraction at longer-time scale. There might be other related studies I missed. I feel that the authors should discuss the relevant findings more thoroughly.

We thank the reviewer for bringing this study to our attention. Upon closer inspection of the study by Dekel and Sagi, 2015, it seems that their finding of 2- to 3-back attraction can be explained by positive response correlations between target trials (Sections 3.4 and 4.4 in Dekel and Sagi, 2015). This might explain why Dekel and Sagi observe a switch from 1-back repulsion to 2- and 3-back attraction, whereas in our perceptual comparison experiment (Experiment 4), there are repulsive biases throughout the sensory history. Please note that our Experiment 4 is not susceptible to similar response correlations, since repeating motor responses does not lead to a systematic bias in our analysis (as there is a more complex relationship between inducer stimulus orientations and answering which of the two comparison stimuli is oriented more clockwise). Conversely, Dekel and Sagi might not observe short-term attraction biases, since they use a 2-AFC paradigm, which minimizes the involvement of working memory, playing an important role in attractive serial dependencies (see Experiment 3; Bliss et al., 2017; also compare to Experiment 4 which uses a 2-AFC paradigm and does not show attractive biases). We now discuss the study by Dekel and Sagi in the Discussion section (subsection “Distinct timescales of attractive and repulsive serial dependencies”).

Reference:Barbosa, João, and Albert Compte. "Build-up of serial dependence in color working memory." bioRxiv (2019): 503185.